# Effects of Pipe Deflection and Arching on Stress Distribution and Lateral Earth Pressure Coefficient in Buried Flexible Pipes

Murat Gulen * and Havvanur Kilic

Department of Civil Engineering, Yildiz Technical University, 34220 İstanbul, Türkiye; kilic@yildiz.edu.tr
* Correspondence: mgulen@yildiz.edu.tr

**Abstract:** In this study, full-scale laboratory tests were conducted on a 315 mm diameter HDPE pipe under shallow buried and localised surface loading conditions to investigate the effects of pipe deflection and arching on stress distribution and the lateral earth pressure coefficient. The tests were validated using 2D finite element software, and further analyses were carried out through parametric studies. These studies considered variations in pipe stiffness, burial depth, backfill properties and pavement stiffness to increase the reliability of the test results. For a shallowly buried HDPE pipe, a comprehensive explanation is provided regarding the evolution of the lateral earth pressure coefficient within the central soil prism. Initially set at $K_o$ conditions, this coefficient tends to shift towards $K_p$ with increasing arching and transitions to $K_a$ with weakening arching. The findings suggest that stress predictions in the crown region of shallow buried flexible pipes are achievable through the application of Terzaghi's arching theory, contingent upon an accurate estimation of the lateral earth pressure coefficient for the central soil prism. Furthermore, the horizontal deflection of the pipe at the springline results in compressive behaviour and passive effects in the surrounding backfill in this specific region. This situation demonstrates that the horizontal stresses at the springline and the lateral earth pressure coefficient can be reliably estimated by considering them as functions of the horizontal deflection of the pipe.

**Keywords:** arcing; earth pressure distribution; lateral earth pressure coefficient; flexible pipe

## 1. Introduction

The variations in pipe diameter are one of the most important performance indicators of buried pipe systems [1,2]. It is therefore, a design criterion of 5% or 7.5% of the pipe diameter is recommended for flexible pipes [3–5]. Notably, Motahari and Abolmaali [6] conducted a comprehensive study using laser scanners and video recordings to assess the deformation characteristics of buried HDPE pipelines spanning a total length of 4572 m across 96 pipelines. Their findings revealed that a substantial majority, 63%, exceeded the deformation limit proposed by AASHTO [5].

In buried pipe systems, the arching phenomenon is employed to reduce stresses acting on the pipe and pipe deflection. Pipe deflection can be defined as either vertical expansion and horizontal shortening or vertical shortening and horizontal expansion of the pipe diameter. To achieve this, researchers recommend the use of different low-stiffness materials (such as EPS) in different buried structures and burial depths, under different loading conditions and in various geometries [7–12]. While these materials significantly reduce stresses and pipe deflection, the resulting surface displacements are larger than of the original case [13]. Tafreshi et al. [14] explored the use of geocell to minimise surface displacements resulting from the use of EPS in shallow buried pipes with successful outcomes. However, they also pointed out that the combined use of EPS and geocell was not economical.

One design option involves predicting new stress distributions at key points of the pipe section due to pipe deflection and incorporate this into the design. In this context,

an understanding the parameters influencing pipe deflection behaviour is essential. In shallow buried flexible pipes, several parameters impact pipe deflection. These include the depth and stiffness of pipe burial, the properties of the backfill surrounding the pipe, the compaction technique and degree, the soil support supplied to the pipe, the location and frequency of load application, the stiffness of the pavement, and the distribution of stresses around the pipe [15–22].

Flexible pipes exhibit either deflection or vertical extension behaviour when loaded. If horizontal stresses ($\sigma_h$) and vertical stresses ($\sigma_v$) around the pipe are equal, variations in the pipe section is unremarkable (Figure 1a). However, if $\sigma_h$ is more dominant than $\sigma_v$, the pipe will extend vertically (Figure 1b). Conversely, when $\sigma_v$ is more dominant than $\sigma_h$, the pipe deforms and expands laterally (Figure 1c). With the deflection of the pipe, some of the load it carries is transferred to the soil prism on the sides of the pipe. This load transfer mechanism involves the pipe displacing more downward than the surrounding soil, leading the central soil prism to move along with the pipe. As the central soil prism tends to move downwards, it interacts with the surrounding soil prisms in this case, facilitating the transfer of some of the load to the adjacent soil through friction forces (Fv) (Figure 1d). This process is a phenomenon known as "positive arching" [23–28].

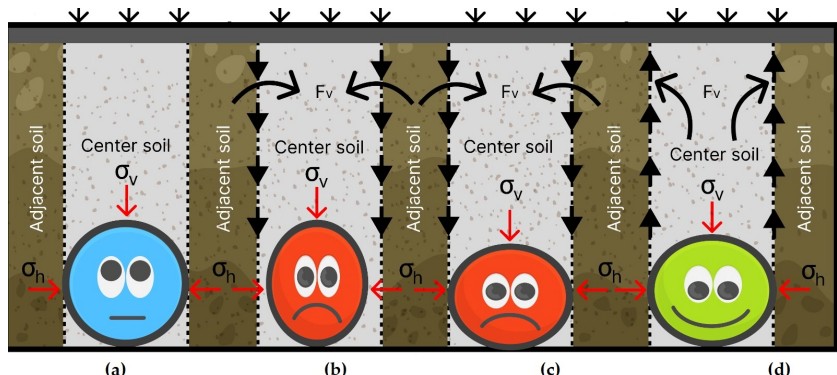

**Figure 1.** Pipe deflection and arching for various stress conditions; (**a**) $\sigma_h = \sigma_v$ (**b**) $\sigma_h > \sigma_v$ (**c**) $\sigma_h < \sigma_v$ (**d**) phenomenon of positive arching.

Terzaghi [25,26] developed a test setup (trapdoor) with a moving door underneath and used it to investigate the effects of soil arching. During the test, the trapdoor was gradually moved downwards. At each stage, both the downward displacement of the trapdoor and the soil loads acting on the trapdoor were measured. Observations showed that the soil load decreased rapidly as the trapdoor moved downwards. Terzaghi [25,26] noted that the soil loads acting on the trapdoor were equal to the difference between the weight of the soil between the shear planes and the Fv generated along these planes. The Fv occurring at a specific height is the basic parameter creating the arching (Figure 1d). Unlike other researchers, surcharge stresses were used as a component of the equation proposed for the arching theory. In addition, the lateral earth pressure coefficient (K) was described as the ratio of horizontal to vertical stresses measured at different points in the region where arching occurred.

The soil arching exists not only in buried pipes but also in various engineering constructions such as culverts, tunnels, piled embankments, retaining walls, and foundations. Researchers have studied the formation, stages, boundaries, and configuration of arching within the soil prism under different testing conditions [29–40].

Arching and deflection of the pipe change the stresses, giving rise to new stress distributions [8,13,41]. This situation results in a significant variation in the lateral earth pressure coefficient, K. In the designing of deep buried flexible pipes, the Iowa equation developed by Spangler [42] and German standards [43] are widely used to calculate uniformly distributed stresses acting on the pipe, assuming equal deflection in both vertical and horizontal directions. Similar approaches are required when considering the distribution of

stresses in shallow buried flexible pipes exposed to localised surface loading. However, there is still a lack of studies on this subject. To develop similar approaches, it is critical to understand the deflection behaviour of the pipe and its effect on the stress distribution, the deformed shape of the pipe and the changes in lateral earth pressure coefficients within the central soil prism due to arching effects.

In this study, two main cases are investigated by full-scale tests and numerical analyses for a shallow buried HDPE (High Density Polyethylene) pipe subjected to localised loading, considering crown and springline stresses and pipe deflections. Firstly, the study explored the applicability of Terzaghi's arching theory to calculate stresses acting on the crown. This involved considering variations in the lateral earth pressure coefficient (K) within the central soil prism, dependent on the form of pipe deflection. In the second case, the changes in springline stresses due to horizontal pipe deflection were studied. Finally, the research investigated the numerical relationship between stresses at the springline and the arching deflection of the pipe through Finite Element (FE) models. This comprehensive approach aimed to provide insights into the complex interaction between soil behaviour, pipe deflection, and stress distributions under localised loading conditions.

## 2. Experimental Setup

The full-scale test system employed for laboratory testing has rigid external walls and a hydraulic loading system (Figure 2). The full-scale tests, plotted to scale in Figure 2a, utilize units in meters (m). The system consists of two types namely; the test without a pipe (S test) and the test with a pipe (P test). In these tests, the load cell type earth pressure cells (L) were employed to measure the stresses around the pipe, while potentiometers were utilised to measure the deflections of the pipe. Base on the pipe's position, the load cells were placed as illustrated in Figure 2. All of the 10 load cells positioned around the pipe were numbered to enhance the clarity of the tex.

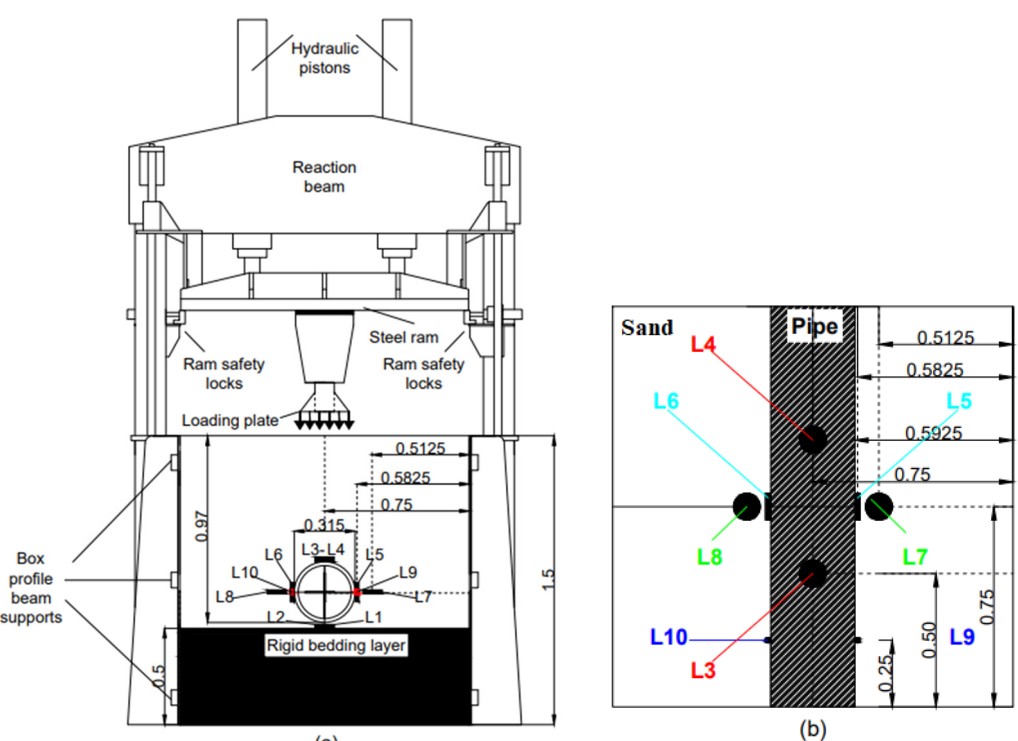

**Figure 2.** Full-scale test system and instrumentation scheme; (**a**) front view and (**b**) top view.

Load cells of L1 and L2, which are not visible in Figure 2b, are located under the pipe and are aligned with L3 and L4.

The load cells used have a maximum operating temperature of 60 °C and a minimum operating temperature of −20 °C. Throughout the tests, the laboratory temperature remained within the operating limits of the load cells, minimising the effect of environmental temperature on their performance. Although the load cells are supplied with a calibration certificate for specific temperature ranges, all of the load cells were calibrated in the laboratory and within the backfill, providing results in line with the values stated on the calibration certificate.

Considering the location of the pipe in Figure 2, four load cells were strategically placed at 0.50 m from the tank wall, where the endpoints of the pipe make contact. These load cells were used to determine the vertical stresses at the crown of the pipe (L3 and L4) and at the invert of the pipe (L1 and L2). These measurements were conducted using TML KDG-500 kPa cells imported from Japan. Furthermore, horizontal soil stresses at the pipe springline (L5 and L6) were determined using TML KDG-200 kPa cells. The cells made direct contact with the pipe wall. Additionally, another set of four cells was used in the soil to record vertical stresses within the pipe zone. The center of two of these cells were located 0.08 m to the right and 0.08 m to the left of the springline (L7 and L8). These measurements were performed using TML KDG-200 kPa cells. The centers of the remaining two cells, which are smaller than the others, are 0.01 m to the right and 0.01 m to the left of the pipe springline (L9 and L10). Moreover, for the purpose of measuring both vertical and horizontal pipe deflections throughout the loading stages, potentiometric displacement sensors were fixed within the pipe at its midpoint section, utilising Burster 8709-5100 sensors manufactured in Germany (Figure 3e). Also, two extra linear variable differential transformers (LVDT) were positioned at the top of the loading plate to measure surface displacement ($S_d$) during both the S and P tests (Figure 3c).

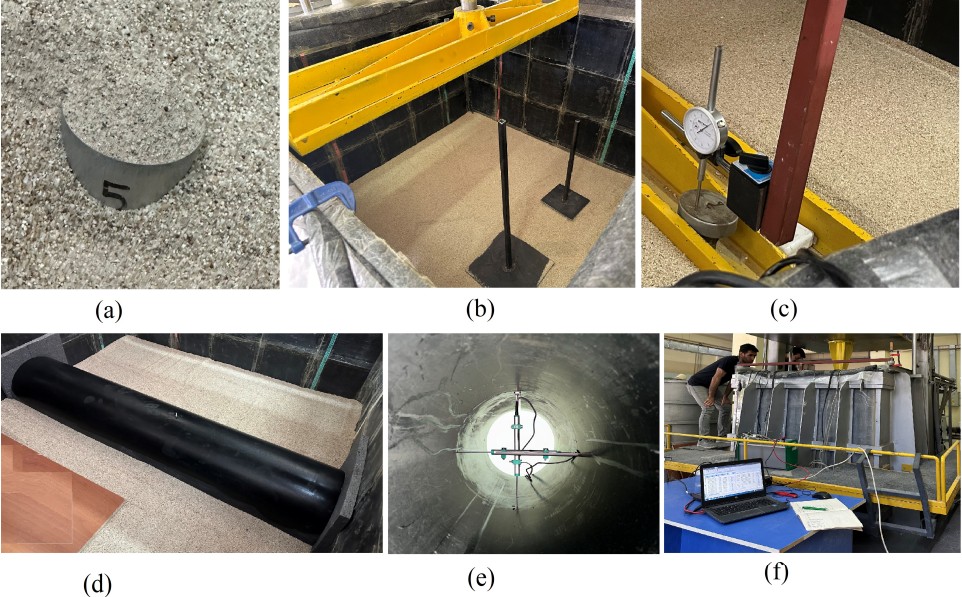

$\qquad$(a)$\qquad\qquad\qquad\qquad$(b)$\qquad\qquad\qquad\qquad$(c)

$\qquad$(d)$\qquad\qquad\qquad\qquad$(e)$\qquad\qquad\qquad\qquad$(f)

**Figure 3.** Stages for full-scale test; (**a**) control of unit weight (**b**) compaction plates and scale (**c**) LVDT (**d**) pipe layout (**e**) potentiometers and (**f**) software and data logger.

In the S and P tests, the backfill was compacted in a controlled manner using two steel plates. The first plate measures 0.15 m × 0.15 m × 0.02 m and weighs 2.5 kg, while the second plate measures 0.30 m × 0.30 m × 0.02 m and weighs 4.5 kg (Figure 3b). The burial depth in both tests is equal to the pipe diameter, D, (0.315 m) and the loads are applied as a strip along the pipe cross-section. This burial depth complies with the minimum of 0.305 m recommended by Watkins and Reeve [17] and Lohnes et al. [18]. The test cross-sections are shown in Figure 3.

### 2.1. Test System

Figure 2 illustrates a full-scale test tank measuring 1.5 m in length, width and height. The top 0.50 m from the bottom comprises rigid steel. To minimise interfacial friction between the tank wall and the soil during the tests, the tank walls were covered with two layers of polyethylene, one layer of geotextile and one layer of protective geomembrane, following the method proposed by Tognon et al. [44]. In addition, the outer membrane surfaces were scaled (Figure 3b). Prior to each test, the geotextiles were lubricated with Dow Corning Molykote 44 High-Temperature Bearing Grease, allowing the system to move freely vertically. This application positively contributed to a reduction of up to 5° in the interfacial friction angle in a large-scale direct shear box. Notably, this method has been used in studies performed by various researchers [45,46].

### 2.2. Materials

In the P test, *a smooth-walled* HDPE pipe 1.42 m in length with an outside diameter of 0.315 m, a wall thickness of 0.02 m, and a density of 970 kg/m$^3$ was utilized. To prevent soil ingress into pipe during the test, the pipe ends were capped with 0.035 m thick XPS25 (Figure 3d). The grain size distribution and physical properties of the backfill, classified as *poorly graded sand* (*SP*) according to the Unified Soil Classification System (USCS), were determined. The optimum water content of 9.8% and the maximum unit dry weight of 17 kN/m$^3$ were derived from standard Proctor tests (Figure 4). Furthermore, the backfill was dry compacted using various methods in a 0.4 m × 0.4 m × 0.4 m square tank, resulting in the maximum unit dry weight values ranging from 16.9 to 17.4 kN/m$^3$.

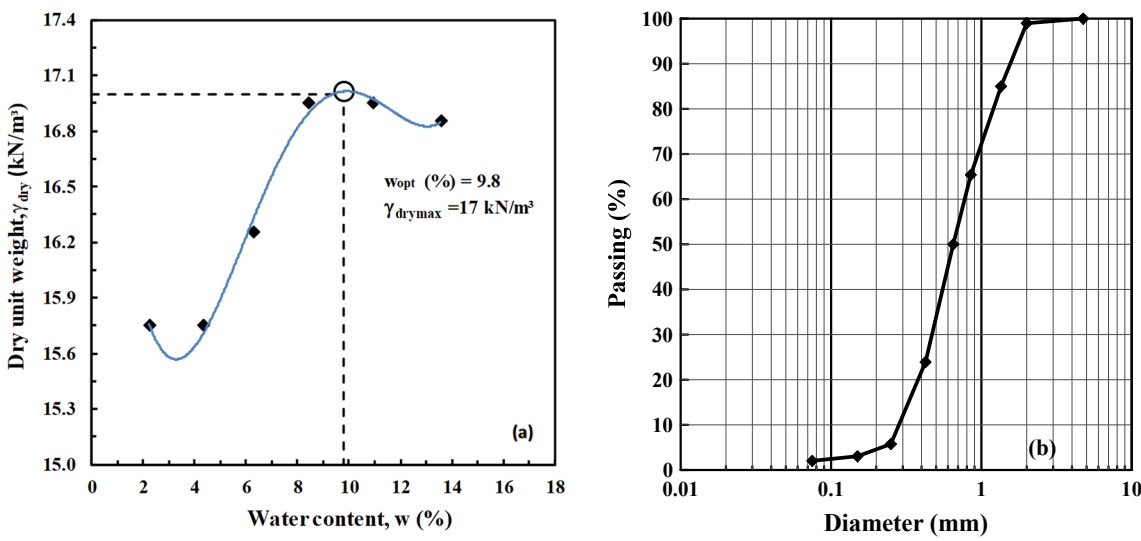

**Figure 4.** Properties for the sand; (**a**) compaction curve and (**b**) grain size distribution.

### 2.3. Loading

To simulate the loads acting on a buried pipe system under strip loading conditions, a 0.05 m thick, 1.45 m length and 0.254 m wide steel plate was placed on the soil surface, as shown in Figure 3b. It was precisely aligned with the mid-section of the pipe. A double-acting hydraulic system connected to a loaded frame was used to apply a vertical load onto the plate (Figure 1). Initially, a force of 22.5 kN (approximately 60 kPa) was applied during the first loading stage. The next stage began once the surface displacement in this first loading stage had terminated. Each stage lasted about 10 min. The subsequent loading stages consisted of forces of 34 kN (90 kPa), 49 kN (130 kPa), 64 kN (170 kPa), 79 kN (210 kPa), 94 kN (250 kPa), 109 kN (290 kPa) and 120 kN (320 kPa), respectively.

*2.4. Test Procedure*

Considering the maximum dry unit weight attained from the standard compaction test, a calculated 190 kg of dry sand is expected to be present at a height of 0.05 m within the 1.48 m × 1.48 m test tank. Therefore, a test backfill mass of 190 kg per layer was adopted. To check the maximum unit dry weight of the backfill, eight moulds, each 0.05 m in diameter and 0.045 m deep, were randomly placed on the surface prior to soil placement. After the tests, the moulds were carefully removed from inside the layer and weighed (Figure 3a). Compaction phases are performed using compaction plates (Figure 3b). The pipe was positioned 0.03 m above the rigid tank bottom to facilitate the placement of L1 and L2 cells in the invert.

**3. Numerical Analysis**

The analyses were carried out using the two-dimensional finite element software PLAXIS (2021) under plane strain conditions to simulate the results of full-scale tests subjected to strip loading and to investigate variations in lateral soil pressures. After validating the finite element (FE) modeling, it was used to conduct a parametric study to investigate factors affecting the proposed system's behaviour. The parametric study took into account different burial depths, pavement stiffness, backfill, and pipe material properties, which have a remarkable effect on pipe deflection and arching. Detailed information on numerical modeling stages is provided below.

*3.1. Mesh and Boundary Conditions*

The general geometry of the model consists of 15-node triangular elements. To assess the accuracy of the analysis, various finite element meshes were generated, including very fine, fine, medium, and coarse elements. It was determined that a medium mesh would suffice. The finite element network created for one-half of the model comprises 1864 elements and 15,487 nodes. Given the symmetry of the two-dimensional model, it was considered adequate to model only half of it to reduce computation and the number of elements required. The actual boundary conditions in the model tests were reflected in the analyses. Since the rigid tank walls and rigid tank bottom in the model tests restricted the horizontal movement of the system, the lateral movement of the model system was fixed. As a result, the system was allowed to move freely in the loading direction and was fully fixed in the opposite direction.

*3.2. Material Properties*

Buried flexible pipes are usually modeled using plate, beam-column, or shell elements [28,47–50]. Most researchers have preferred to use elastic material models, generally employing the secant modulus of elasticity to simplify the analyses and avoid complexity [13,47,48,51]. Researchers utilising an elastic material model for the pipe typically employ the short-term secant modulus of elasticity, commonly set at 450 MPa.

The plate elements in numerical models are treated similarly to conventional beam elements, including shear, bending, and normal forces in their formulation. This study modeled HDPE pipe and steel loading plate using plate elements assuming elastic behaviour and considering axial and bending stiffness. The modulus of elasticity for HDPE pipe is 450 MPa, the axial stiffness (EA) value is 9000 kN/m, and the bending stiffness (EI) value is 0.3 $kN/m^2/m$; the modulus of elasticity of steel is used as 200,000 MPa. Table 1 summarizes the parameters used.

Additionally, by compacting the backfill in the large shear box at the maximum dry unit weight, the internal friction angle ($\phi$) and the soil-pipe interface friction angle ($\delta$) were obtained. The modulus of elasticity ($E_{50}$) was also taken from the stress-strain curves of the triaxial compression (CD) tests (Table 1).

**Table 1.** Parameters used in Plaxis 2D analysis.

| Backfill Soil Properties | | | | | | | | |
|---|---|---|---|---|---|---|---|---|
| Unit Weight (kN/m$^3$) | Model | ν | c (kPa) | φ (°) | $E_{50}$ (MPa) | $E_{oed}$ (MPa) | $E_{ur}$ (MPa) | Rint-1 tank-soil |
| 17 | H.S | 0.33 | - | 40 | 28 | 28 | 70 | 0.17 |
| Pipe Properties | | | | | | | | |
| Density (kg/m$^3$) | Model | ν | E (MPa) | EA (kN/m) | | EI (kN/m$^2$/m) | | Rint-2 pipe-soil |
| 970 | L.E | 0.45 | 450 | 9000 | | 0.30 | | 0.50 |

Note: H.S, Hardening soil; L.E, lineer elastic; E, modulus of elasticity; $E_{50}$, secant elasticity modulus; $E_{oed}$, tangent oedometric modulus; $E_{ur}$, unloading-reloading modulus; Rint, coefficient of interaction.

*3.3. Consruction Stages*

Measurements taken by potentiometers positioned inside the pipe during the compaction of the backfill revealed that both horizontal and vertical deflections of the pipe were negligible. Therefore, the installation phases are not detailed in the 2D analyses. The initial phase, referred to in the analyses as the Ko condition (coefficient of earth pressure at rest), signifies the state of readiness for loading. The loading phases were carried out in eight different phases. The capacity of the model test system and the degree of surface displacement played a crucial role in determining the magnitude of the applied loads. The IMC Studio software recorded the earth pressure cell and potentiometer readings for each loading step. Surface displacements at each loading step were measured by LVDTs placed on the steel plate.

**4. Results and Discussion**

In the tests, two measurements were obtained from symmetrically placed load cells and LVDTs for the same measurement point. These two measurements are averaged and presented as a single value when presenting the outcomes.

*4.1. Surface Displacement*

The surface displacement value occurring in the S and P tests consists of 2 components, as expressed in Equation (1).

$$\sum S_d = u_{ys} + u_{ypv} \tag{1}$$

where, $S_d$ surface displacement; $u_{ys}$ displacement of backfill; $u_{ypv}$ pipe vertical deflection

The surface displacement values derived from the S test were utilized in analyses to verify and calibrate the material parameters of the backfill, aligning with findings from the experimental studies.

The $E_{50}$ of the backfill was computed from triaxial (CD) tests to be 24 MPa. However, the best agreement between the surface displacements obtained from the analyses and the test surface displacements was achieved by using the backfill material parameters in Table 1. These values were also used in the analyses for the P tests. Figure 5 presents the variation in surface displacements resulting from both the S and P tests.

The surface displacement value for the S test is 8.76 mm, while for the P test it is 10.71 mm due to the 320 kPa surcharge stress. The data derived from the S and P tests show close agreement, with the surface displacement values obtained from the P test generally being greater than those found for the S test.

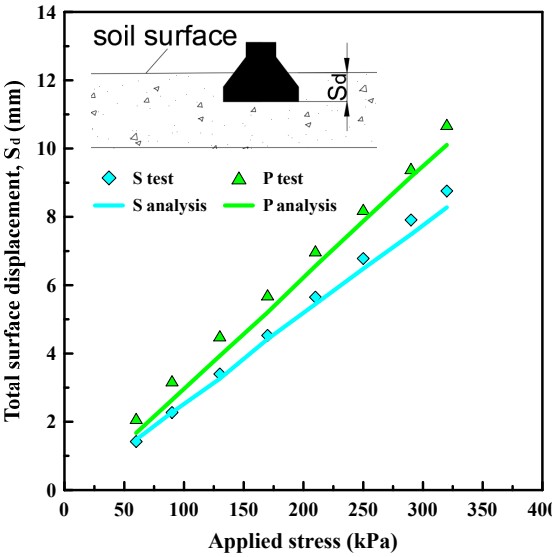

**Figure 5.** The variation of measured and computed surface displacements with applied stress.

*4.2. Vertical Stresses*

During the S and P tests, crown stresses ($\sigma_{crown}$) were measured at L3 and L4, and invert stresses ($\sigma_{invert}$) were measured at L1 and L2. Figure 6 compares these stress values with the results obtained from analysis.

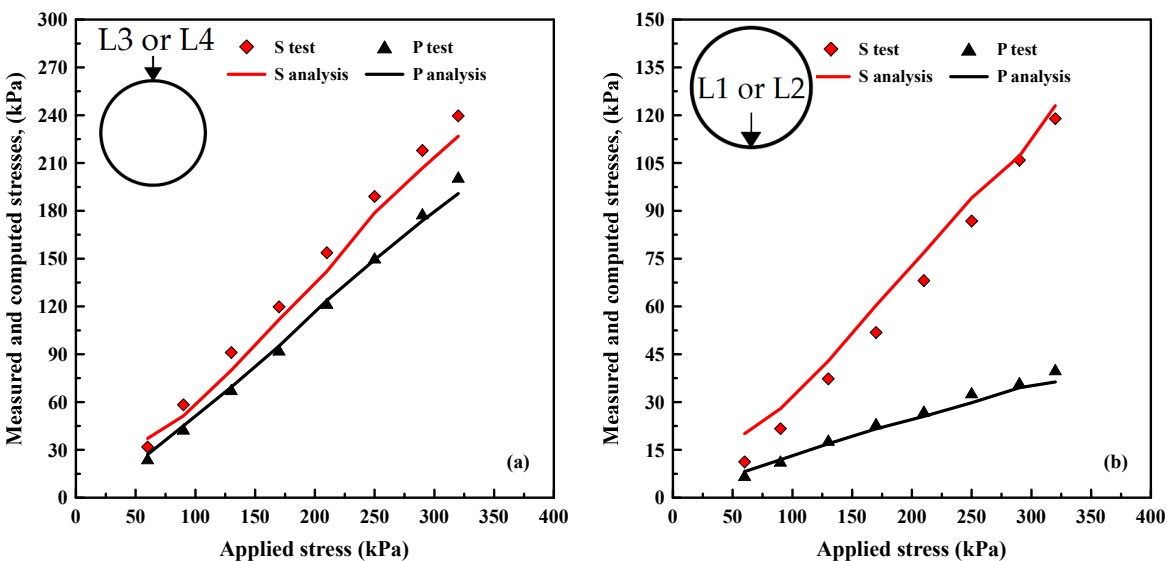

**Figure 6.** The variation of measured and computed stresses for; (**a**) crown and (**b**) invert.

The analyses reveal that in the S test under localised loading conditions, stresses and displacements are distributed over a wider area from the bottom of the loading plate downwards. Different settlements with unclear boundaries occur between the central and adjacent soil prisms beneath the loading plate. The resulting Fv values from these distinct displacement conditions contribute to the arching to a limited extent in the S test. Conversely, in test P, using an HDPE pipe, which leads to more displacement than in the soil, enhances the Fv under boundary conditions with a width approximately equal to the pipe diameter. This positive effect improves the arching and reduces crown stresses.

The vertical stresses acting on the pipe cause pipe deflection and arching. These stresses are transferred to the adjacent soil prism, clearly reducing the invert stresses (Figure 6b). The analyses indicated that the stiffness of the bottom of the pipe and the distribution of stresses over a larger area at the bottom of the pipe contribute to this result.

The $u_{ypv}$ component of the surface displacement in the P test functions in the same way as the trapdoor and is the most critical parameter influencing arching. When $z = 0$ and $h = q$ are adopted in the differential solution of Terzaghi's arching theory equation, Equation (2) is derived. The first part of the equation represents the stresses due to the soil pressure, and the second part represents the stresses due to the surcharge.

$$V = \frac{B(\gamma)}{K\tan(\phi)}\left(1 - e^{-K\tan(\phi)h/B}\right) + q_{sur}e^{-K\tan(\phi)h/B} \tag{2}$$

where, V is the vertical stress after arching, B is the half-width of the moving segment, h is the soil height above the moving segment, q is surcharge stress.

In Terzaghi's arching theory, a clear understanding of the lateral earth pressure co-efficient (K), describing the horizontal effect of V, is crucial. Equation (2) is employed to calculate the vertical stresses at the crown in the P test, with K becoming the sole variable. The study established the variation of $\sigma_{crown}$ using Equation (2) for test condi-tions. Figure 7 compares these stresses to the $\sigma_{crown}$ values derived from the analyses. Figure 7b shows that the stresses determined from the test and analysis correspond to different K values at various loading levels. In particular, K and $\sigma_{crown}$ were determined assuming that the frictional force occurs in vertical planes (Figure 1). The variation of K and $\sigma_{crown}$ can be studied under different assumptions regarding the formation, boundaries, and configuration of the arching in the central soil prism.

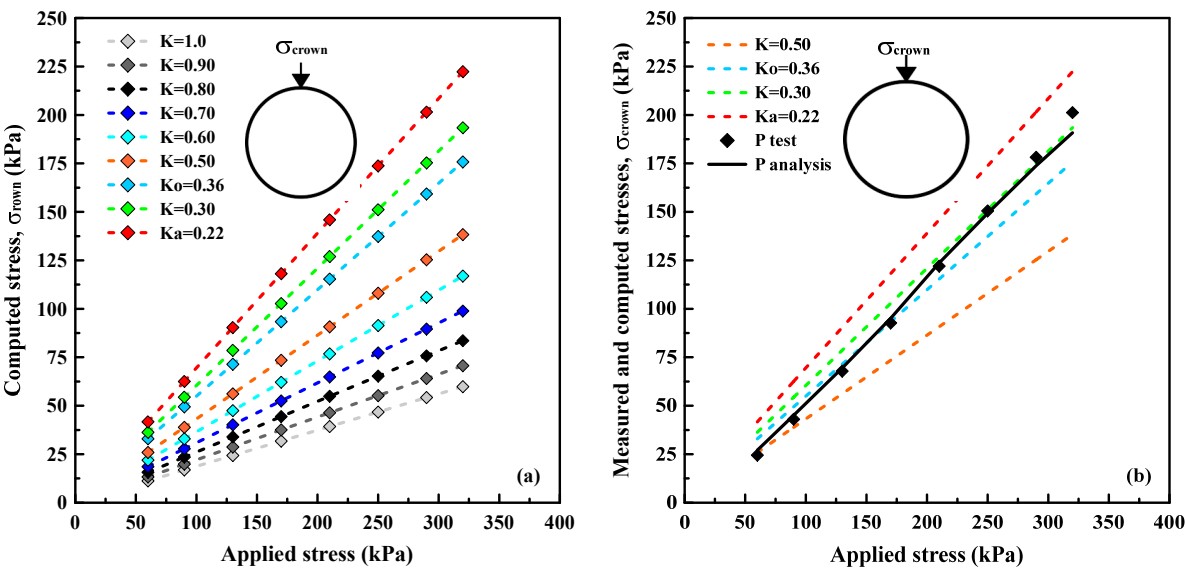

**Figure 7.** Stresses for $\sigma_{crown}$ (**a**) Equation (2) results for different K values (**b**) comparison of stresses.

The enhancement of arching due to the vertical deflection of the pipe increases the stresses transmitted from the central soil prism to the adjacent soil prisms. The shear planes in the soil-pipe friction also significantly aid in the transfer of vertical stresses to the sides of the pipe. This increased stress transfer to the adjacent soil prism results in higher stresses in the P test compared to the S test (Figure 8).

The vertical stresses transmitted to the adjacent soil prism due to arching, as well as the vertical effect of the stresses applied from the surface, decrease from the pipe springline towards the tank wall. It is noted that the stresses at points L9 and L10 are approximately 45% lower than the stresses at points L7 and L8 (Figure 8). This rate of reduction particularly increases from the springline towards the tank walls.

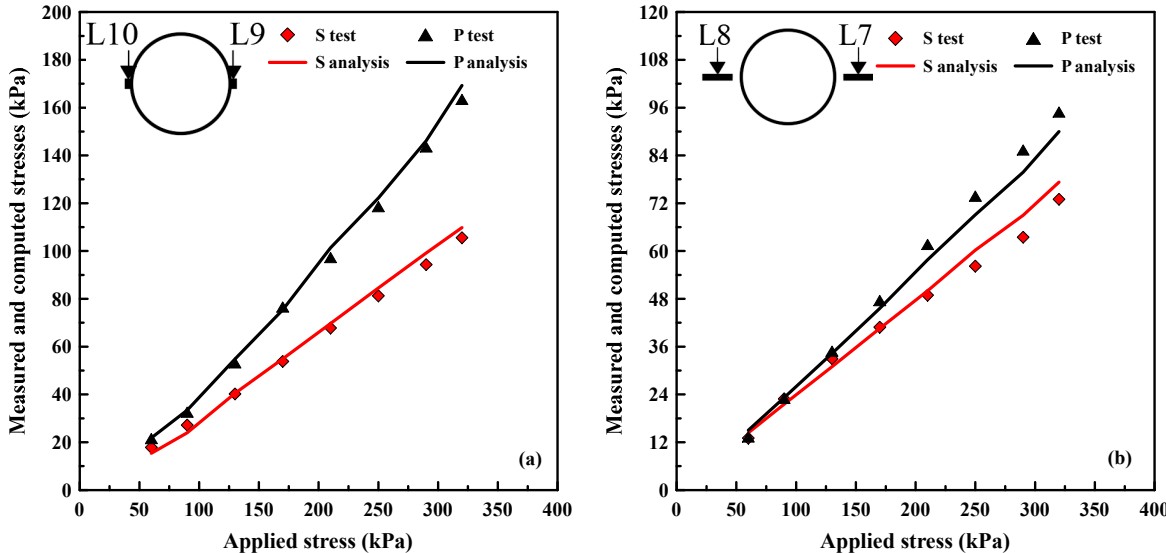

**Figure 8.** Vertical stresses for adjacent soil prism; (**a**) L9 and L10, (**b**) L7 and L8.

### 4.3. Horizontal Stresses

Horizontal stresses are essential for defining the lateral earth pressure coefficient at the springline. In both the S and P tests, L5 and L6 cells were positioned at the location of the pipe springline. Figure 9 illustrates the variation in horizontal stresses measured in the tests and computed from the analyses.

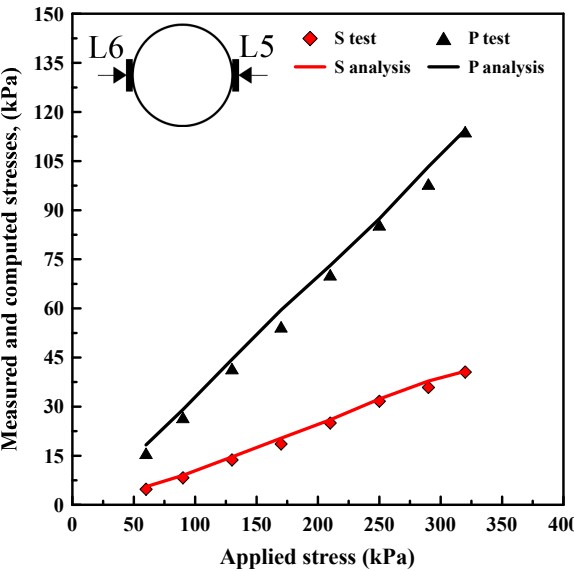

**Figure 9.** Variation of horizontal stresses in the springline.

In the P test, a horizontal deflection of about 5 mm was recorded at the springline. This deflection causes the soil particles in the springline zone, initially in the Ko condition, to move towards the tank wall.

Substantial variations in stresses were observed at the same locations in the S and P tests, considering arching, pipe deflection, and stress distribution. As the crown (L3–L4) and invert (L1–L2) stresses decreased, the stresses at L9–L10 and L7–L8 increased due to arching. The effect of vertical stresses transferred to the adjacent soil prism is clearly evident at points L9–L10, while this effect is somewhat reduced at points L7–L8. The most significant alteration, mainly attributed to the influence of horizontal deflection (Ph), was observed in the horizontal stresses at the springline.

### 4.4. Pipe Deflection

The stress distribution and vertical deflection of the HDPE pipe under localised loading conditions without pavement on the topsoil surface are shown in Figure 10.

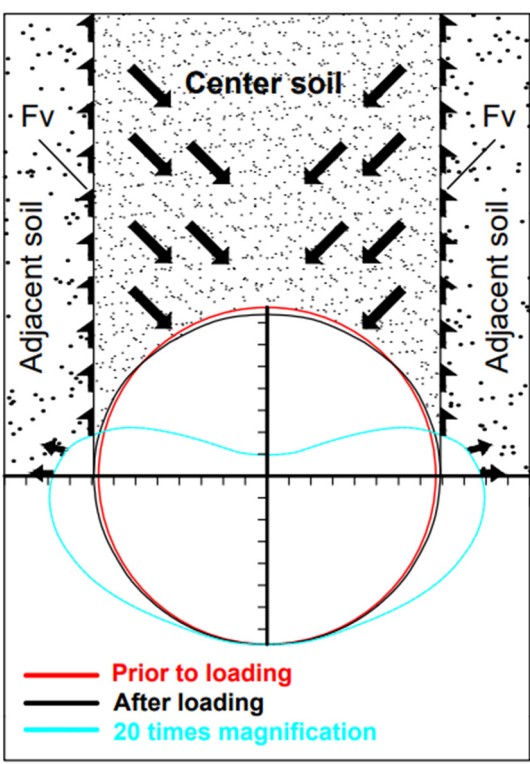

**Figure 10.** Distribution of stress and deflection of pipe subjected to localised loading.

The analysis indicates that the vertical deflections of the pipe are non-uniform and concentrated at the crown, depending on the stress distribution. The heart-shaped deflection pattern emerges as a consequence of the crown descending more than other parts of the top of the pipe. This distinct deformation is a characteristic outcome of the specific deflection behaviour observed in the system (Figure 10). In addition to the vertical deflection of the flexible pipe, the applied stresses also cause horizontal deflection in the pipe. This has a significant effect on the behaviour of the backfill at the sidewalls of the pipe.

Ph reaches a maximum value approximately at the springline. Considering the deformation form of the pipe, the vertical deflection of the pipe (Pv) is greater than that of Ph. In addition, the ratio of the horizontal deflection of the pipe to the vertical deflection (Ph/Pv) is expressed as the deformation ratio ($d_r$). The deformation ratio ranges from 0.76 to 0.87, with a mean of 0.81 for the test, and from 0.76 to 0.80, with a mean of 0.79 for the analysis (Figure 11). Although the $d_r$ for the analysis is consistent, the $d_r$ obtained for the test data fluctuates at high stress levels.

Ph is the primary parameter determining the horizontal stresses in the springline. Similarly, Pv increases the vertical stresses transmitted to the sides of the pipe by enhancing the arching effect. Therefore, a relationship exists between the pipe deflections and the stresses determined at the sides of the pipe. To clarify this relationship further, the measurement points where L9 and L10 are located are denoted as SP-1. The variations of horizontal stresses ($\sigma_{h1}$) at SP-1 as a function of Ph and the variations of vertical stresses ($\sigma_{v1}$) at SP-1 as a function of Pv are illustrated in Figure 12.

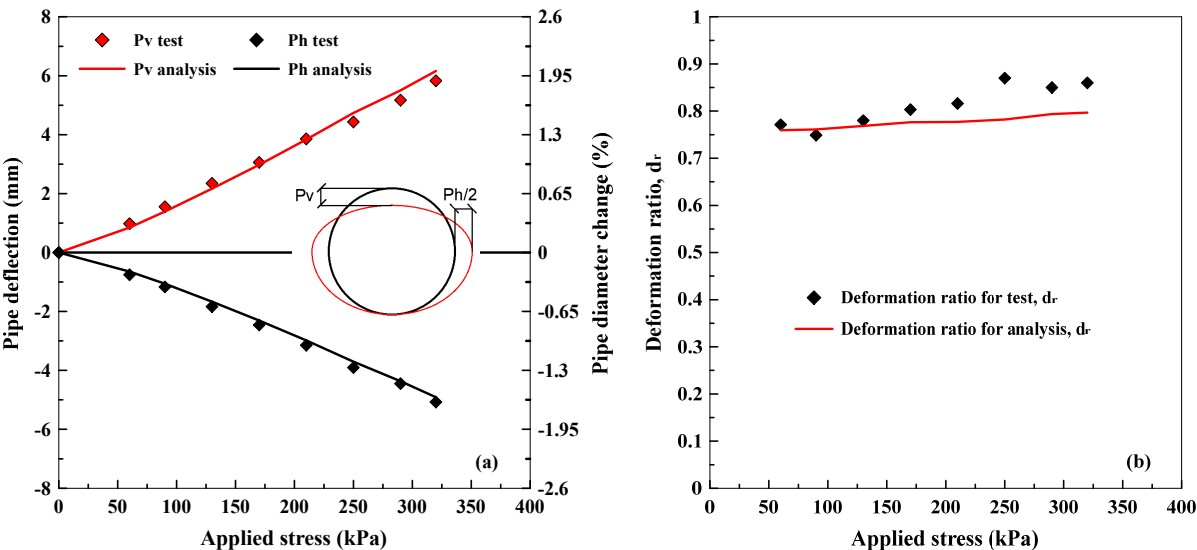

**Figure 11.** Pipe deflection versus stress levels (**a**) Pv and Ph variation (**b**) variation of $d_r$.

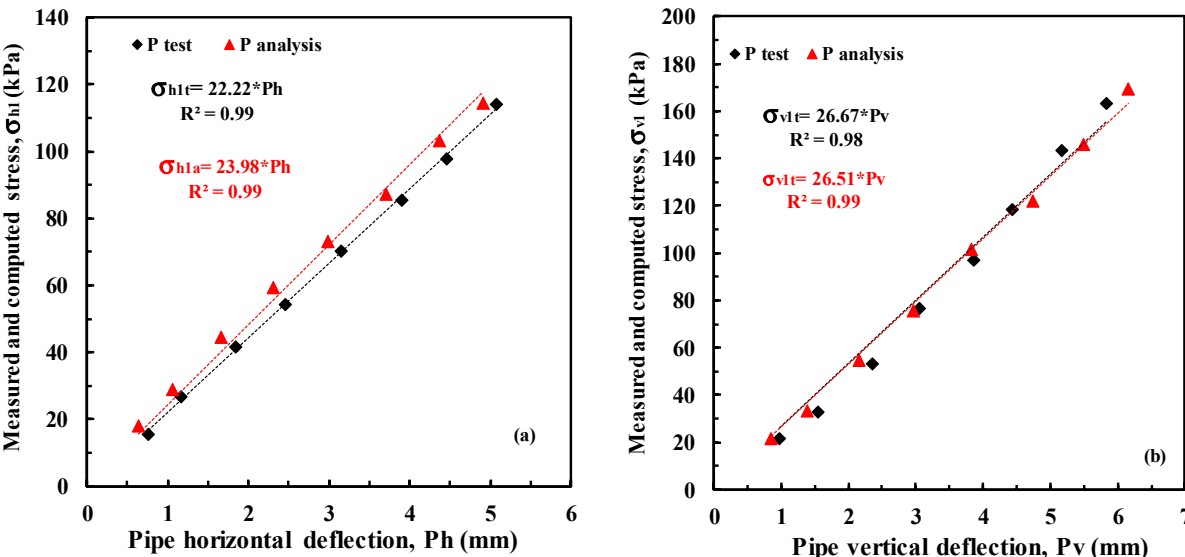

**Figure 12.** Variation of stresses at the SP-1 with respect to pipe deflection (**a**) Ph and $\sigma_{h1}$ (**b**) Pv and $\sigma_{v1}$.

The relationship between Ph and $\sigma_{h1}$ at SP-1 is defined by Equation (3) for both test and analysis, while Equation (4) describes the relationship between Pv and $\sigma_{v1}$. However, the preferred backfill and pipe properties for buried pipe systems vary widely depending on the application. Therefore, the relationships presented in this study are valid for the loading conditions, backfill, and pipe properties.

$$\sigma_{h1t} = 22.22 \times Ph_t \tag{3a}$$

$$\sigma_{h1a} = 23.98 \times Ph_a \tag{3b}$$

$$\sigma_{v1t} = 26.67 \times Pv_t \tag{4a}$$

$$\sigma_{v1a} = 26.51 \times Pv_a \tag{4b}$$

In Equations (3) and (4), $\sigma_{h1t}$, $\sigma_{v1t}$, $Ph_t$ and $Pv_t$ signify the measured values in the test, while $\sigma_{h1a}$, $\sigma_{v1a}$, $Ph_a$ and $Pv_a$ signify the values computed from the analysis.

### 4.5. Change of Lateral Earth Pressure Coefficient

Terzaghi [25,26] proposed a value of 1 for K in the region where arching occurred. It was asserted that K varies between Ko and 1.5 depending on the vertical movement of the trapdoor and the soil height at which the frictional force appears. Researchers investigating arching mechanisms in various structures have reported K values for arching formations ranging from Ka (Rankine's active lateral earth pressure coefficient) to 1.6 [23,30,33–36,52–56]. Lin et al. [39] explain arching in tunnels and claim that the lateral earth pressure coefficient in the central soil prism varies from Ko to Kp (coefficient of passive lateral earth pressure) depending on the burial depth, which is not constant.

When comparing the stresses computed for different values of K in Equation (2) with the test and analysis outcomes, two distinct situations emerge. In the first four loading stages, where Pv is low and more homogeneously distributed, the soil behaves similarly to the Ko condition due to limited downward movement of the central soil prism. However, as surcharge stresses increase, stress concentration occurs predominantly in the crown, resulting in deflections that take on a heart-shaped form. The movement of particles towards the crown in the central prism is caused by intensified surcharge stresses, specific to shallow buried flexible pipes, negatively affecting the Fv by creating arching. Consequently, the lateral earth pressure coefficient is reduced, reaching a value between Ko and Ka (K = 0.3) (Figure 7). In other words, the greater the Pv in the crown of the pipe compared to other regions, the closer the earth pressure coefficient tends to be to Ka.

The K value at SP-1 was computed for both S and P tests using Equation (5). Substituting $\sigma_{h1}$ and $\sigma_{v1}$ from Equations (3) and (4) into Equation (5) generates Equations (6) and (7).

$$K = \frac{\sigma_{h1}}{\sigma_{v1}} \tag{5}$$

$$K_t = \frac{22.22 \times Ph_t}{26.67 \times Pv_t} = c_1 d_r \tag{6}$$

$$K_a = \frac{23.98 \times Ph_a}{26.51 \times Pv_a} = c_1 d_r \tag{7}$$

where, $c_1$ represents the coefficient that characterises the correlation between pipe deflections and stresses in the adjacent soil prism, and the lateral earth pressure coefficient is defined for Ka analysis computations for Kt test measurements

For SP-1, the variation of K was obtained from Equation (5) and the variation of K was computed as a function of $d_r$ in Equations (6) and (7) for testing and analysis (Figure 13).

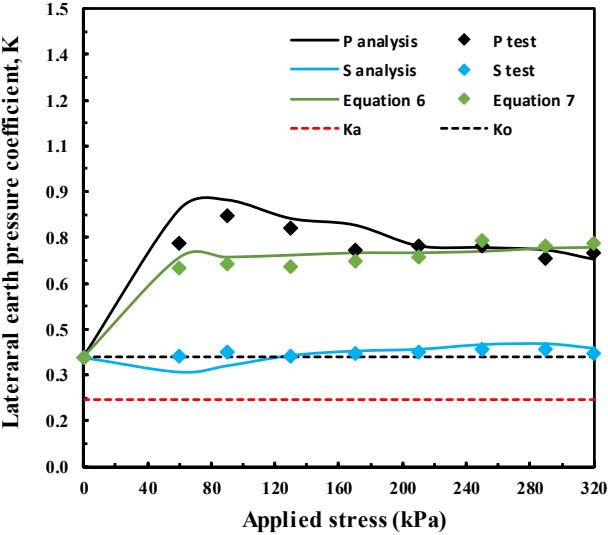

**Figure 13.** Variation of the K with applied stress.

In the S test and analysis, it was observed that the maximum horizontal stresses at the spring line were approximately 40 kPa, resulting in a horizontal displacement of 0.85 mm. This limited movement of the backfill towards the tank wall was influenced by the horizontal stresses. However, it did not have a significant effect on the backfill, which initially exhibited a Ko state.

Considering the first four loading stages in both the P test and the analysis, the increment of horizontal stresses at low Ph values is greater than the increment of vertical stresses transferred to the adjacent soil prism by arching. This leads to higher K values in the first loading stages. However, after approximately 2–2.5 mm of horizontal pipe deflection, the increments for both horizontal and vertical stresses stabilize at SP-1, which means that K is constant. Although the K values obtained at low stresses for the P test are initially lower than those found in the analyses, they become more consistent as the stresses increase and are not affected by the applied stresses

For the first four loading stages, the K values computed based on $d_r$ and $c_1$ in Equations (6) and (7) are lower than the K values derived from testing and analysis according to the ratio of horizontal to vertical stresses ($\sigma_{h1}/\sigma_{v1}$), and the K values are more consistent with increasing stress levels.

### 4.6. Results of the Parametric Study

The model and boundary conditions used to verify the test data were retained in the parametric studies. In the analyses, all parameters except the variable of interest were kept constant.

### 4.6.1. Effect of Rigid Pavement

In shallow buried pipe systems exposed to localised surface loading, the stiffness of the pavement and its presence are two critical factors affecting pipe performance and serviceability. The distribution of stresses applied from the surface is significantly influenced by the stiffness of the pavement under load [13]. To investigate this effect, an infinitely thin and weightless rigid plate was defined along the soil surface under load, without altering the height of the backfill, the magnitude of the applied stresses or the impact area. The results of the analyses with and without pavement were compared using the stress variation ratio ($r_{vs}$) in Equation (8) and the deformation variation ratio ($r_{vd}$) in Equation (9) (Figure 14). The terms di and $d_f$ in the equations refer to displacement or deflection, while the terms $\sigma_i$ and $\sigma_f$ represent stresses. To compute the rates of variation, the values of $\sigma_i$ and $d_i$ were used for the test condition and the values of $\sigma_f$ and $d_f$ were used for a rigid pavement. Measurement points L7 and L8, specifically to investigate the change in horizontal and vertical stresses from the springline to the tank walls, are referred to as SP-2. For this region, vertical stresses are expressed as $\sigma_{v2}$ and horizontal stresses as $\sigma_{h2}$.

$$r_{vs}(\%) = \frac{\sigma_i - \sigma_f}{\sigma_i} \tag{8}$$

$$r_{vd}(\%) = \frac{d_i - d_f}{d_i} \tag{9}$$

The presence of a rigid plate allowed the surface stresses to be distributed over a larger and more uniform area. Due to this circumstance, maintaining the applied load constant while increasing the impact area resulted in significant changes in the measurements. These changes led to a reduction of 80% in the stresses, 84% in the $S_d$ and 82% in the Pv and Ph. In particular, the reduction in the Pv, which is one of the main causes of arching, significantly reduced the transfer of load from the central prism to the adjacent prisms.

The rigid plate provided a stress distribution close to a uniformly distributed load condition. The stress and stress distribution varies slightly with depth, resulting in surface stresses acting on the crown. This behaviour renders Terzaghi's arching theory dysfunctional in Equation (2).

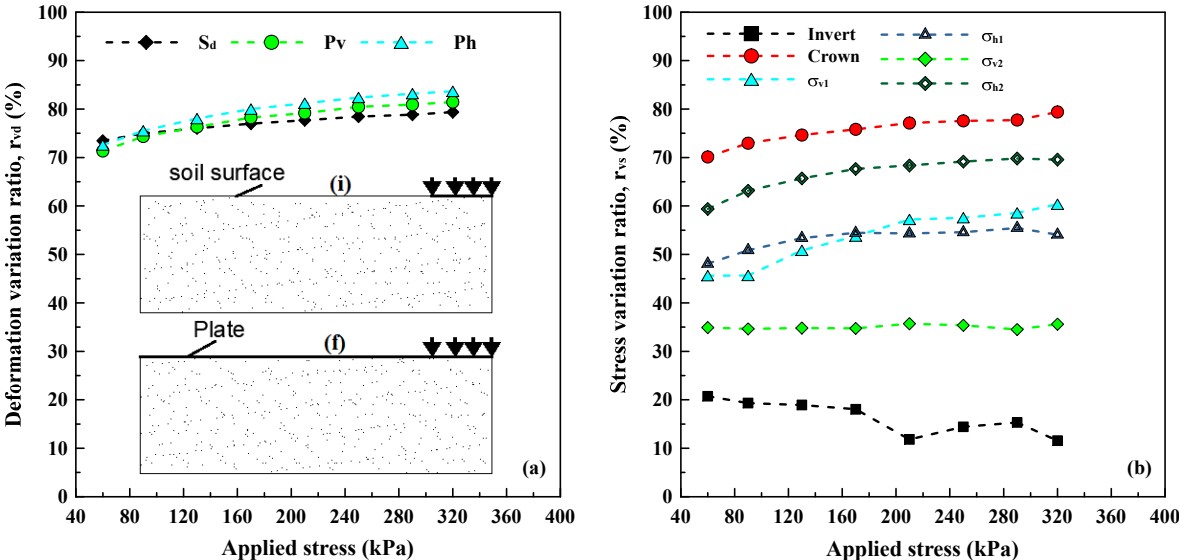

**Figure 14.** The variation of stress and deformation for rigid pavement; (**a**) $r_{vd}$ (**b**) $r_{vs}$.

The 82% reduction in horizontal pipe deflection significantly reduces $\sigma_{h1}$ and the horizontal incident distances of this value. When there is no rigid plate, $\sigma_{h1}$ decreases by about 30% close to the tank wall, while in the case of a rigid plate, its effect diminishes after 0.5D from the springline. As it approaches the tank wall from the SP-1 point, the lateral stresses lose their effect, causing K to tend towards Ko. In the final situation, the changes in Ph and Pv in Figure 14a are parallel to the changes in $\sigma_{v1}$ and $\sigma_{h1}$ in Figure 14b.

The variation in the values of $\sigma_{h1}$ and $\sigma_{v1}$, determining K, is similar as seen in Figure 14b. Therefore, the value of K in SP-1 for the pavement use case exhibited significantly less variation at different loading levels compared to Figure 13. Furthermore, changes in $c_1$ and $d_r$ in Equation (7) were observed when a rigid plate was present on the soil surface. The $d_r$ decreased compared to the condition without a plate and showed a limited change in the range of 0.69–0.72. The $c_1$ increased to reach 1.07. Thus, the increase in one component of K and the decrease in the other component did not lead to considerable changes in the K values.

4.6.2. Effect of Burial Depth

Elevating the burial depth (h) increases the $u_{ys}$ component and decreases the $u_{ypv}$ component as a consequence of lowering the stresses acting on the pipe. Although the change in $S_d$ in the final state was minor, the proportion of $u_{ypv}$ in $S_d$ had a considerable effect on the arching.

As a result of the decrease in locally applied stress from the surface soil with increasing burial depth, remarkable variations in the stresses acting on the pipe or culvert were observed depending on, the burial depth [13,21,28]. Stress concentration in the crown region and the corresponding Pv tend to direct particle movement within the central soil prism from the Ko state to the Ka state. Figure 15 illustrates the variation of stresses determined from Equation (2) and the analysis for the crown.

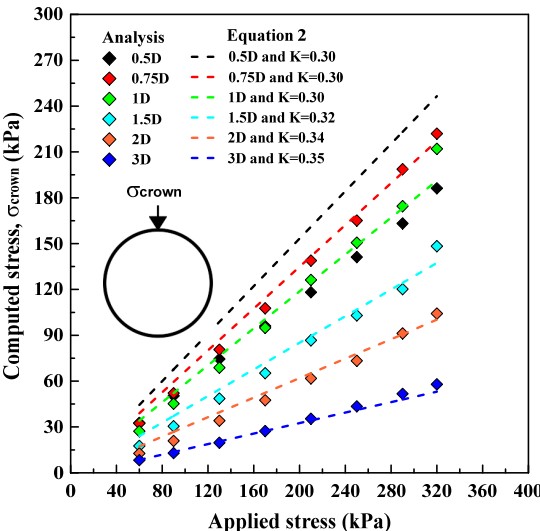

**Figure 15.** Comparison of stresses obtained from Equation (2) and analysis for crown.

The analyses revealed that when h was 0.5D, both Pv and arching indicated local arching in the crown, leading to a significant reduction in stresses within the crown region. This resulted in the maximum stress occurring at a certain distance from the crown. However, Equation (2) does not take into account this specific behaviour of shallow buried flexible pipes, making the data incompatible at a burial depth of 0.5D.

The stress variation at the crown of the pipe significantly affects both flexing and arching. The values of $r_{vs}$ and $r_{vd}$ were determined for burial depths ranging from 0.5D to 3D, with reference to a burial depth of 1D (Figure 16). These $r_{vs}$ and $r_{vd}$ remained approximately the same for different loading levels at a constant burial depth, allowing comparisons to be made by averaging these values for a general assessment.

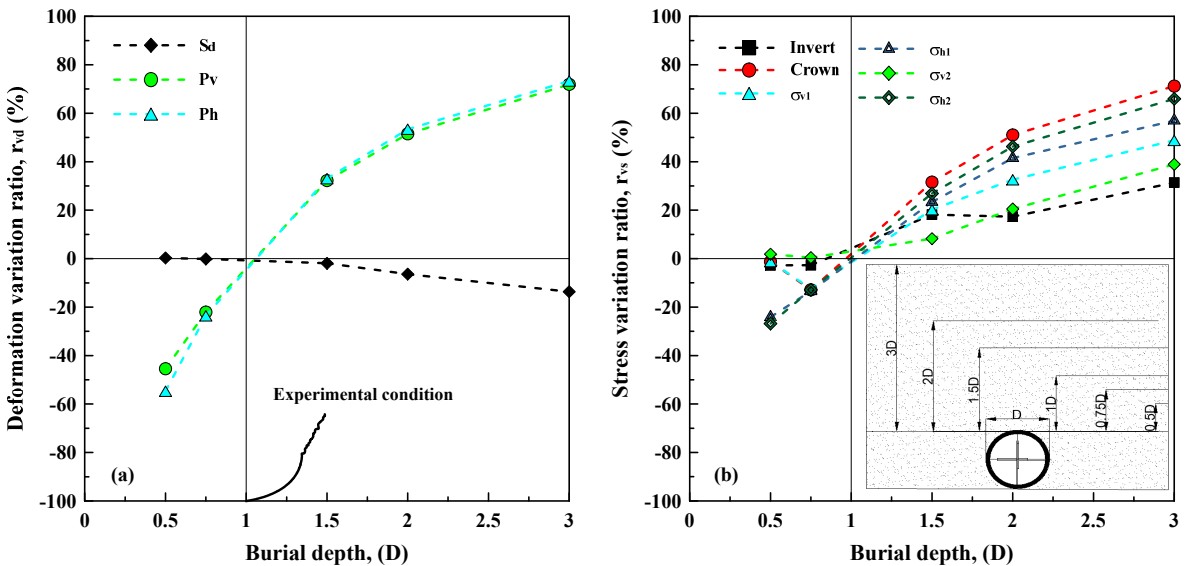

**Figure 16.** Stress and deformation variation ratios for different h values (**a**) $r_{vd}$ (**b**) $r_{vs}$.

Increasing the burial depth resulted in a decrease in $\sigma_{h1}$, but did not significantly alter the distance over which $\sigma_{h1}$ was effective. Similarly, Pv and stress transfer to the sides of the pipe decreased as the burial depth increased. The decrease in $\sigma_{h1}$ was more pronounced than the decrease in $\sigma_{v1}$. Therefore, the lateral soil pressure coefficient at SP-1 decreased with increasing burial depth.

The increase in burial depth positively affected the $d_r$ values, which ranged from a mean of 0.74 to 0.83. The values for $c_1$ at different burial depths were 0.845, 0.891, 0.90, 0.915, 0.875, and 0.845. Figure 17a demonstrates the variation of K for different loading levels and burial depths. Figure 17b shows the K obtained from Equation (7) using $c_1$ and the mean $d_r$, as well as the change in K computed from the $\sigma_{v1}/\sigma_{h1}$ ratio for the analyses.

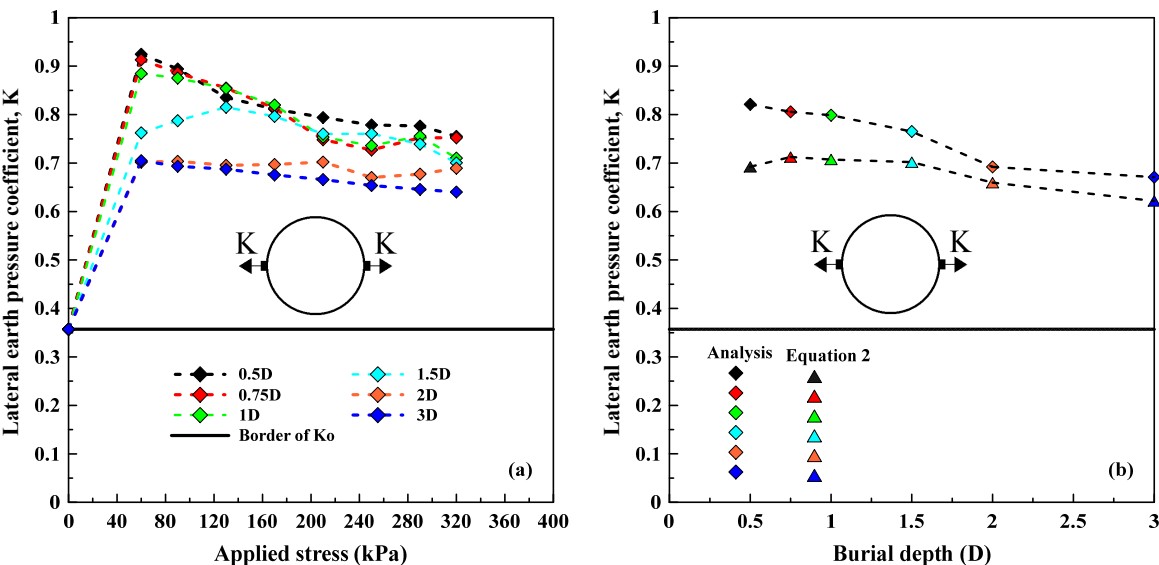

**Figure 17.** Variation of K at SP-1 with regard to h (**a**) K for different stresses (**b**) mean of K values.

### 4.6.3. Effect of the Stiffness of the Pipe and Backfill

The analyses were repeated while keeping the backfill modulus of elasticity (Es) constant and considering a range of pipe modulus of elasticity (Ep) from 450 MPa to 2800 MPa. The increment in Ep significantly reduces the $u_{ypv}$ component in $S_d$, while the increment in stresses acting on the central soil prism resulted in a limited increase in the $u_{ys}$ component. In particular, the reduction in $u_{ypv}$ prevents the formation of a heart formed deformation.

The stresses applied to the buried pipe system are distributed based on the relative stiffness of the pipe and the surrounding backfill. An increase in pipe stiffness amplifies the stresses carried by the pipe [19,28]. Additionally, a reduction in Pv decreases the stresses transmitted to the adjacent soil prism. The decrease in K implies a reduction in the horizontal effect of vertical stresses in the central soil prism. Figure 18 illustrates the fluctuation in stresses derived from the analyses and Equation (2) for various stiffness ratios.

Increasing the Ep leads to a significant reduction in the horizontal and vertical pipe deflection. This reduction in Ph results in a substantial decrease in $\sigma_{h1}$. These stresses become negligible at a distance of D/4. Between D/4 and 1.5D, the horizontal component of the stresses applied from the surface remains effective, but its effectiveness decreases significantly beyond this distance. The gradual decline in $\sigma_{v1}$ corresponds to the disappearance of the arching. The decrease in horizontal stress is more pronounced close to the tank wall than the decrease in vertical stress. Moreover, increasing the modulus of elasticity (Ep) not only restrains pipe deflection but also facilitates stress transmission from the crown to the invert.

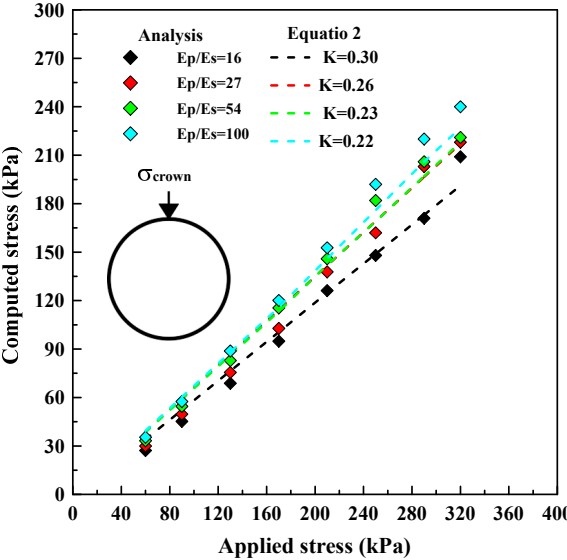

**Figure 18.** Variation of $\sigma_{crown}$ values.

The $r_{vs}$ and $r_{vd}$ values were computed for various Ep/Es ratios, using the stress and deflection or displacement obtained with an Ep value of 450 MPa as a reference (Figure 19). The term stiffness ratio ($r_s$) is defined to express these different Ep/Es ratios.

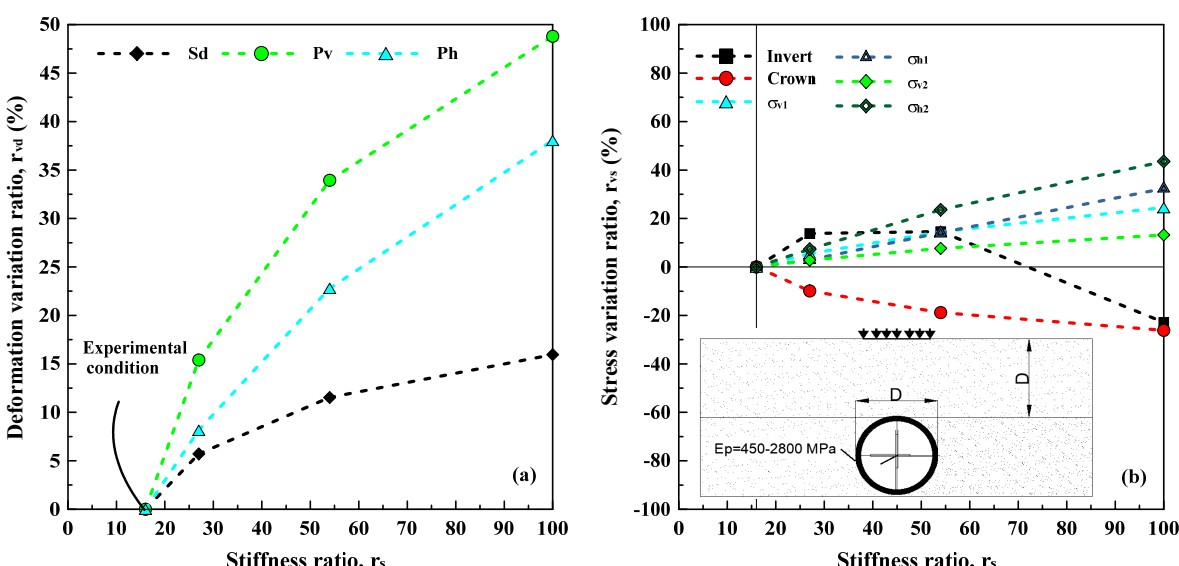

**Figure 19.** Stress and deformation variation ratios for different $r_s$ (**a**) $r_{vd}$ (**b**) $r_{vs}$.

The heart-shaped deformation pattern was gradually disappeared as Ep increased, resulting in a more uniform distribution of deflections at the top of the pipe. This led to a greater decrease in Pv than in Ph, and the $d_r$ gradually approached 1. For $r_s$ values of 16, 27, 54, and 100, the $d_r$ values were determined to be 0.79, 0.85, 0.91, and 0.94, respectively. The coefficient $c_1$ for K in Equation (7) was computed as 0.90, 0.85, 0.78, and 0.73 based on the $r_s$ values. The change in K for different stress levels and $r_s$ values was derived from the analyses and Equation (7) (Figure 20).

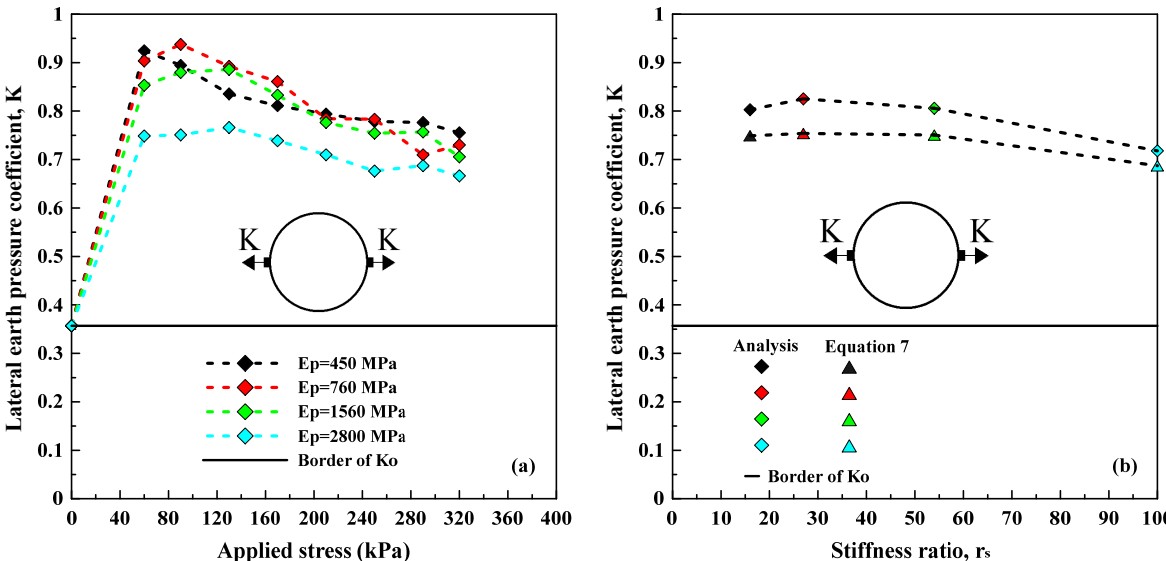

**Figure 20.** Variation of K concerning Ep for SP-1 (**a**) for different stresses (**b**) mean K for different $r_s$.

The gradual increase in the stiffness of the backfill from 28 MPa to 115 MPa, while keeping the Ep constant, leads to a decrease in the $u_{ys}$. Similarly, the reduction in the stresses carried by the pipe causes to a reduction in the $u_{ypv}$ component. The higher stiffness of the backfill surrounding the pipe enhances the soil support provided to the pipe by the backfill [28,57]. The soil support enables a significant reduction in pipe deflections. Generally, the changes in the system resulting from the increase in pipe stiffness for arching are reversed by the increase in backfill stiffness. The $r_{vs}$ and $r_{vd}$ values were computed for various $r_s$ with reference to the Es value of 28 MPa (Figure 21).

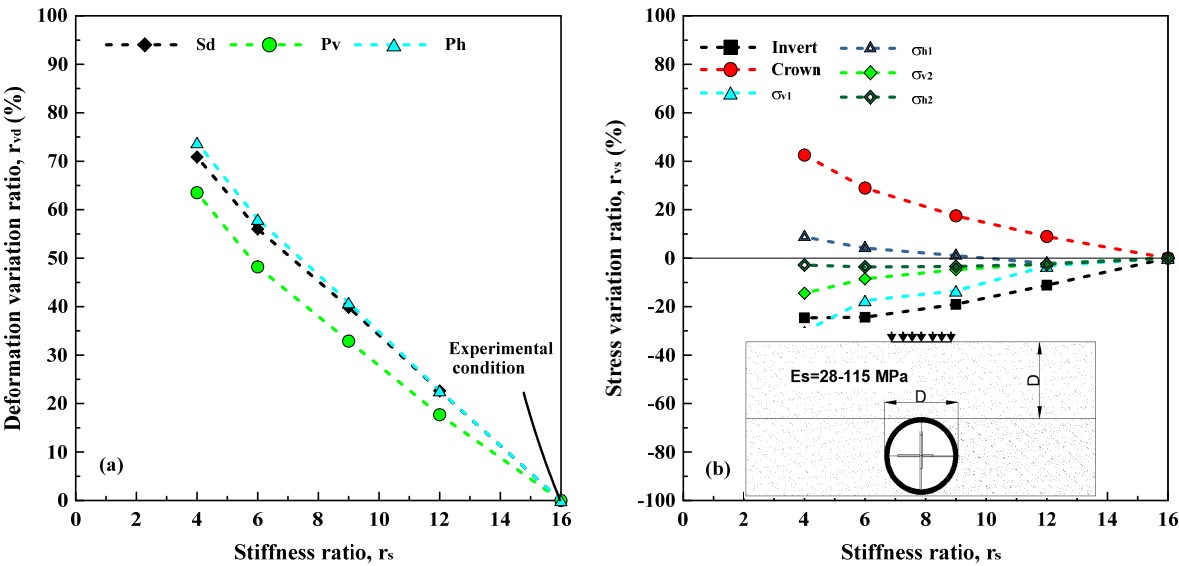

**Figure 21.** Stress and deformation variation ratios for different $r_s$ (**a**) $r_{vd}$ (**b**) $r_{vs}$.

Increasing the stiffness of the backfill enhances Fv [14,58], amplifying the horizontal stress component acting on the central soil prism. In other words, the lateral earth pressure coefficient, which represents the horizontal effect of the vertical stresses, also increases with Fv. Figure 22 shows the stresses in the crown obtained using Equation (2) and the analyses.

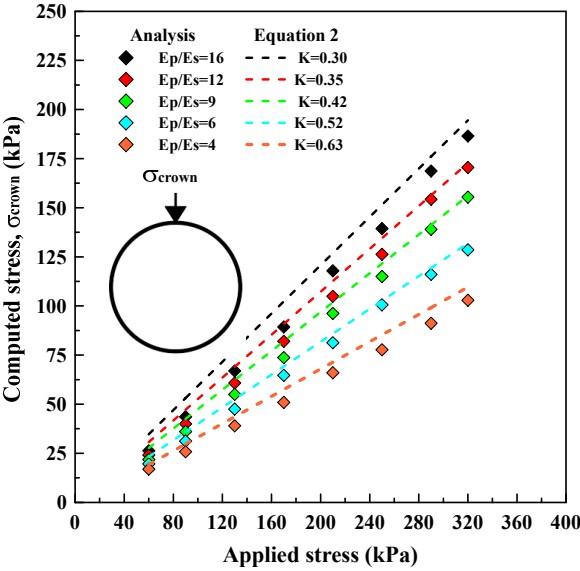

**Figure 22.** Variation of $\sigma_{crown}$ for different $r_s$.

When Es was raised, there was a notable reduction in both Ph and Pv. However, this resulted in only a marginal reduction of up to 10% in $\sigma_{h1}$, which extended to the tank wall. In contrast, $\sigma_{v1}$ increased by 30% due to the more efficient working of the arching, resulting in a decrease in K.

The mean $d_r$ values for stiffness ratios of 16, 12, 9, 6, and 4 were 0.79, 0.73, 0.69, 0.63, and 0.56, respectively. The corresponding $c_1$ coefficients for these values were 0.97, 1, 1.02, 1.05, and 1.07. Figure 23 illustrates the variation of the K values derived from both analyses and Equation (7). Increasing the stiffness of the backfill significantly reduces K at low stress levels, and this effect gradually diminishes with increasing stresses (Figure 23a).

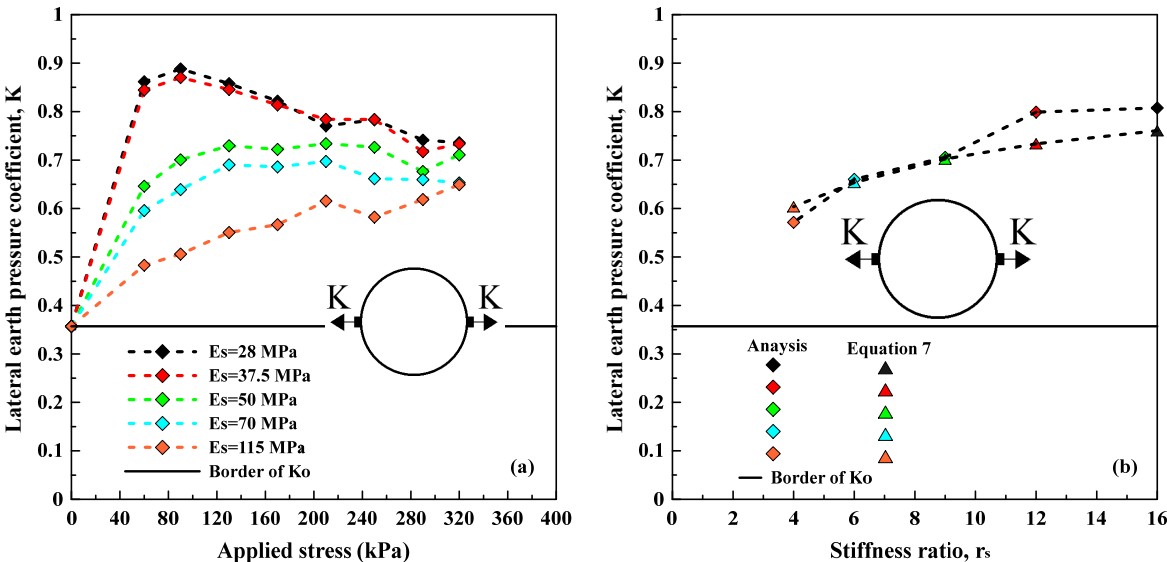

**Figure 23.** Variation of K for SP-1 (**a**) for different stress values (**b**) for different $r_s$.

## 5. Conclusions

The various internal and external loads acting on shallowly buried flexible pipe systems cause horizontal and vertical deflection of the pipe. These deflections and arching results stress distributions and lateral earth pressure coefficients that differ from the initial conditions in the buried pipe system. This study investigates the effects of pipe deflection and arching on stress distributions and lateral earth pressure coefficients through full-scale

laboratory experiments, analysis and proposed empirical equations. The findings are presented below.

1.  Pipe deflections are a key parameter defining the behaviour of the system, particularly in the crown and springline regions. The lateral earth pressure in the Ko condition tends towards Kp as the arching effect increases in the central soil prism, and towards Ka as the arching effect weakens. The correct estimation of K in the central soil prism in the final situation indicates that the stresses applied to the crown can be computed using Terzaghi's arching theory.

2.  The horizontal deflection of the pipe causes the backfill at the springline to move in the direction of the tank wall. As a result, the lateral earth pressure coefficient tends to convert from Ko to Kp, leading to an increase in horizontal stresses. This variation in K increases further towards the tank wall. In buried flexible pipe designs, where pipes are spaced and placed side by side, these horizontal effects will impose additional stresses on the neighbouring pipe. It is therefore essential that these effects are considered at the design stage.

3.  The correlation between pipe deflections and stresses at the SP-1 was observed to be strong ($R^2 = 0.99$). This relationship indicated that both horizontal and vertical stresses in the pipe springline can be defined by estimating pipe deflections.

4.  The presence of a rigid pavement in shallow flexible buried pipes provided a more homogeneous distribution of localised stresses over a wider area. This variation in stress distribution resulted in a reduction in stress and deflection of up to 84%. Additionally, the lateral earth pressure coefficient approached Ko beyond a distance of D/3. This transition occurred due to the significant reduction in horizontal stresses from the spring line to the tank wall.

5.  Increasing the burial depth led to a significant reduction in stresses and pipe deflections. Specifically, the horizontal stresses at the springline decreased more than the vertical stresses, and the effect of these horizontal stresses extended to the tank wall.

6.  The increase in pipe stiffness caused a noteworthy reduction in pipe deflections and horizontal stresses at the springline. In particular, the reduction in $\sigma_{v1}$ was more pronounced than the reduction in $\sigma_{h1}$ at the springline, and the gradual increase in Ep led to the establishment of the Ko condition along the sides of the pipe.

7.  The increment in backfill stiffness considerably reduced both Ph and Pv, enhancing soil support and arching. However, there was no considerable variation in the lateral stresses transmitted from the pipe to the backfill, and in the influence distance of these stresses, as the pipe stiffness remained constant.

The results obtained from the tests, analyses and empirical equations in this study are valid for shallow buried flexible HDPE pipes subjected to the surcharge stresses, test procedures, material parameters and local loading considered. To achieve a more comprehensive evaluation, further experimental studies are necessary.

**Author Contributions:** Conceptualization, M.G. and H.K.; methodology, M.G. and H.K.; software, M.G. and H.K.; validation, M.G. and H.K.; investigation, M.G.; original draft preparation, M.G.; review and editing, M.G. All authors have read and agreed to the published version of the manuscript.

**Funding:** This study was conducted with financial support from Yıldız Technical University (YTU) BAP Coordination Unit under the FBA-2022-5124 and FBA-2022-4576 projects.

**Data Availability Statement:** The data presented in this study are available on request from the corresponding author (privacy).

**Acknowledgments:** We would like to express our gratitude for the financial support provided by Yıldız Technical University (YTU) BAP Coordination Department under the FBA-2022-5124 and FBA-2022-4576 projects. We appreciate the support from Yıldız Technical University (YTU) BAP Coordination Department. We would like to express our sincere gratitude to Özgür Yıldız and Serhan Ulukaya, a faculty members at Yıldız Technical University, for their invaluable assistance in facilitating the publication of this study and for their unwavering support throughout the process.

**Conflicts of Interest:** The authors declare no conflict of interest.

**Abbreviations and Notation**

D: Diameter

$d_i$: initial displacement or deflection

$d_f$: Final displacement or deflection

$d_r$: Deformation ratio

EA: Axial stiffness

EI: Bending stiffness

Es: Soil elasticity modulus

Ep: Pipe elasticity modulus

EPS: Expanded Polystyrene

FE: Finite element

Fv: The frictional force in vertical planes

h: Burial depth

HDPE: High Density Polyethylene

H.S: Hardening soil

K: Lateral earth pressure coefficient

Ka: Rankine's active lateral earth pressure coefficient

Ko: Lateral earth pressure coefficient at rest

Kp: Passive lateral earth pressure coefficient

L: Loadcell

L.E: Linear elastic

LVDT: linear variable differential transformer

m: meter

SP: Poorly graded sand

Ph: Pipe horizontal deflection

Pv: Pipe vertical deflection

$r_s$: Stiffness ratio

$r_{vd}$: Deformation variation ratio

$r_{vs}$: Sress variation ratio

$S_d$: Surface displacement

SP: Poorly graded sand

SP-1: Measuring location 1 in the springline

SP-2: Measuring location 2 in the springline

$u_{ypv}$: Pipe vertical deflection

$u_{ys}$: Displacement of backfill

$\sigma$: Stress

$\sigma_{crown}$: Stress in crown

$\sigma_f$: Final stress

$\sigma_h$: Horizontal stress

$\sigma_{h1}$: Horizontal stress in SP-1

$\sigma_{h2}$: Horizontal stress in SP-2

$\sigma_i$: Initial stress

$\sigma_v$: Vertical stress

$\sigma_{v1}$: Vertical stress in SP-1

$\sigma_{v2}$: Vertical stress in SP-1

USCS:Unified Soil Classification System

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
