# Peer review of "Effects of Pipe Deflection and Arching on Stress Distribution and Lateral Earth Pressure Coefficient in Buried Flexible Pipes"

_applsci, doi:10.3390/app14041667_

Round 1

Reviewer 1 Report

Comments and Suggestions for Authors

The article relates to full-scale laboratory tests on a 315 mm diameter pipe working under shallow buried and localised surface loading. The authors investigated the effects of pipe deflection and arching on stress distribution and the lateral earth pressure coefficient in the crown and springline regions. They have used the Terzaghi's arching theory to predict some stresses in the crown zone of the shallow buried flexible pipes. Thus, the horizontal stresses in the springline and the lateral earth pressure coefficient were  effectively estimated as functions of the horizontal pipe deflection.

My comments are as follows:

1.Abstract contains abbreviations such as HDPE, Ko, Ka and Kp that need to be explained before or during the process.

2.What does the letter W means in Figure 2?

3.LFDT has not been defined – Page 4, Figure 3.

4.During the tests, the load is almost point-like, Figure 2, while in the ground the load is on the entire surface of the pipe side. Maybe the laboratory conditions are different from the real conditions of a pipe under the ground?

5.What are the following tests: the test without a pipe (S test) and the test with a pipe (P test)? – Page 3. Is there an additional pipe section? Is the S test with only a soil ballast? Why is such research? Is the effect of soil load on the pipe or the effect of the pipe on soil displacement examined?

6.Figure 4b – What is the grain size distribution?  For a grain diameter of 1 m it is 65% why?

7.Line 171 - It should be: The…

8.The created finite element model and its load and supporting condition should be presented? – to explain the description from Lines: .183-185

9.It would be good to present the model used in the tests and simulations on the hardening soil stress-strain diagram. - Table 1.

10.What is the q surcharge stress Line 270? (dot at the end)

11.Let's define and describe the K and Ka parameters - Figure 7, Lines: 271-275, 398-399 etc. They were defined but only in Lines 365-367.

12.Complete a section describing abbreviations.

13.Lines 345-346 blank fragments.

14.The authors write: “This study investigates the effects of pipe deflections and arching on stress distributions and lateral earth pressure coefficients…” but they did not address the effects of pipe diameter or curvature. Research on one type of pipe and one type of soil is a bit limited.

15.The terms rvs and rvd we defined I (8) and (9), but term rs was not defined (Figure 19).

16.Line 512, why the author use the rs (with subscript) and dr (without subscript)?

17.Too many different conclusions. Some are less important - they can be thrown away.

Comments on the Quality of English Language

English language requires improvement.

Author Response

Manuscript ID: applsci-2862955

Murat Gulen1, Havvanur Kilic2

1 Department of Civil Engineering, Yildiz Technical University, [email protected]

2 Department of Civil Engineering, Yildiz Technical University, [email protected]

We sincerely appreciate the time and effort you invested in reviewing our manuscript. It's an honor for us to have the opportunity to submit a revised version. Your feedback and insightful comments have been invaluable to us. We have successfully incorporated changes to address most of the suggestions provided, and these revisions have been clearly highlighted within the manuscript.

Below, we have provided a detailed point-by-point response to your comments and concerns, with corresponding revisions highlighted in red within the resubmitted files. We believe these revisions adequately address your concerns and align with the standards of the journal

Once again, we extend our gratitude for your valuable feedback and constructive criticism.

Murat GULEN (Phd.Candidate)

Yildiz Technical University

Department of Civil Engineering

Reviewer-1

Comment 1. Abstract contains abbreviations such as HDPE, Ko, Ka and Kp that need to be explained before or during the process

Response 1. Thank you for your careful review and valuable feedback. We have added explanations for the abbreviations HDPE, Ko, Ka, and Kp at their first mention after the abstract.

HDPE: High Density Polyethylene (line 95)

Ko: coefficient of earth pressure at rest (line 248)

Ka: Rankine's active lateral earth pressure coefficient (line 392)

Kp: coefficient of passive lateral earth pressure (line 86)

Comment 2. What does the letter W means in Figure 2?

Response 2. We initially tried to use directional indicators such as east, west, south and north to indicate the load cell locations. However, we decided to take a different approach and used Figure 2b to illustrate the load cell locations. In this figure the label 'W' represents the first letter of the word 'West', indicating the load cell location. We apologise for any confusion caused by this oversight. The letter 'W' has been removed from Figure 2 and the figure revised accordingly.

Comment 3. LFDT has not been defined – Page 4, Figure 3.

Response 3. We have responded on the assumption that by the term 'LFDT' you meant 'LVDT'. Thank you for bringing this to our attention. LVDT stands for 'Linear Variable Differential Transformer' and we have clearly defined this term before Figure 3. (Line 141).

Comment 4. During the tests, the load is almost point-like, Figure 2, while in the ground the load is on the entire surface of the pipe side. Maybe the laboratory conditions are different from the real conditions of a pipe under the ground?

Response 4. Thank you for your valuable contribution. The loading conditions in both the tests and analyses were the same, with the loads uniformly distributed over a localised area of 0.254m x 1.45m, as shown in Figure 3a. To further clarify the loading conditions, revisions have been made to Figure 2. Specifically, Figure 2a, Figure 3b and Figure 14a show that the loading conditions were similar throughout the experimental study.

Comment 5. What are the following tests: the test without a pipe (S test) and the test with a pipe (P test)? – Page 3. Is there an additional pipe section? Is the S test with only a soil ballast? Why is such research? Is the effect of soil load on the pipe or the effect of the pipe on soil displacement examined?

Response 5.  The study included only two tests: the S test and the P test. No additional tests or pipe sections were included beyond these two tests. The S test was conducted in a full-scale test tank without a pipe, following the procedure outlined in Section 2.4 and the loading steps described in Section 2.3. During this test, all load cells shown in Figure 2 were positioned in the same location for stress measurements. The P test followed the same procedure and loading steps as described in Sections 2.3 and 2.4. This test was carried out in a full scale test tank using a pipe as shown in Figure 2a.

One of the purposes of the S test, as described in Part 4.1 and in the line 267-271, is to calibrate the parameters of the backfill material. In addition, another important objective of the S test is to evaluate the effect of the pipe deformations observed during the P test on the stress distribution and arching. To evaluate this change due to pipe deformation, it is necessary to compare the results of the P test with those of the S test. Therefore, the S test was performed.

Comment 6. Figure 4b – What is the grain size distribution?  For a grain diameter of 1 m it is 65% why?

Response 6. Thank you for clarifying the unit of the grain diameters in Figure 4b. It has been determined that the correct unit for the grain diameters on the x-axis should be millimeter (mm), not meter (m). We appreciate the opportunity to correct this oversight.

In Figure 4b, the term 'grain size distribution' refers to the distribution of particle sizes within a given material. It is usually expressed as a percentage of particles falling within certain size ranges. For example, a grain size distribution of 65% means that 65% of the particles in the material are 1 mm in diameter.

Comment 7. Line 171 - It should be: The…

Response 7. Thank you for pointing out the error. In line 171, the sentence now begins with "The..." (Line 201)

Comment 8. The created finite element model and its load and supporting condition should be presented? – to explain the description from Lines: .183-185

Response 8. The explanation provided in lines 183-185 is necessary to comprehend the finite element model constructed and its loading and support conditions. This information is critical to understanding both the methodology of finite element analysis and how the structural behaviour of the system is evaluated.

Comment 9. It would be good to present the model used in the tests and simulations on the hardening soil stress-strain diagram. - Table 1.

Response 9. Thank you for your feedback on improving the clarity and visual presentation of the text. The manuscript is currently 24 pages in total and contains 23 figures. We sincerely apologise for not being able to include the requested figures. However, details of these figures can be found below.

Figure 1a illustrates the typical calculation steps for the stress-strain relationship and modulus of elasticity for the hardening soil model. Figure 1b depicts the results of the triaxial (CD) test carried out under laboratory conditions at different confined pressures. Figure 1c shows the calculation steps for the E50 value derived from the triaxial test conducted at 100 kPa confined pressure.            

Comment 10. What is the q surcharge stress Line 270? (dot at the end)

Response 10. Thank you for your question. The 'surcharge stress, q' referred to in line 270 indicates the stress from loads applied from the soil surface. The 'q' values referred to in this study refer to the stresses applied by the loading plate from the top of the soil. These details of the stresses applied by the loading plate are provided in Section 2.3, lines 181-185.

Comment 11. Let's define and describe the K and Ka parameters - Figure 7, Lines: 271-275, 398-399 etc. They were defined but only in Lines 365-367.

Response 11. Thank you for highlighting this point, which we had initially overlooked. In order to provide a clear definition and description of the parameters K, Ko, Kp and Ka, which are referred to throughout the text, we have added a detailed description.

Comment 12. Complete a section describing abbreviations.

Response 12. Thank you for your feedback. In consideration of your comments, a list of abbreviations has been added at the end of the article to improve readability and simplify the presentation of information.

Comment 13. Lines 345-346 blank fragments.

Response 13. Thank you for bringing this oversight to our attention. Blank fragments were removed from lines 345-346.

Comment 14. The authors write: “This study investigates the effects of pipe deflections and arching on stress distributions and lateral earth pressure coefficients…” but they did not address the effects of pipe diameter or curvature. Research on one type of pipe and one type of soil is a bit limited.

Response 14. Thank you for your feedback. Pipe deflection, represented by vertical shortening (Pv) and horizontal expansion (Ph) of the pipe, was investigated throughout the study. Specifically, the effects of pipe deflection on stress distributions and lateral earth pressure coefficients were evaluated using parameters such as Pv, Ph, the Ph/Pv ratio (dr) and Equation 9. These evaluations are clearly shown in Figures 11, 12, 14a, 16a, 19a and 21a. To avoid confusion, the definition of pipe deflection is given in the second paragraph of the introduction (line 35-37).

Throughout the text, we have attempted to present the details of how stresses are transferred from the central soil prism to the adjacent soil prisms by defining the arching phenomenon as illustrated in Figure 1. In particular, the stress variation ratio (rvs) is identified as the most important indicator of the change in stress due to arching. A detailed study of this situation is depicted in Figures 14b, 16b, 19b and 21b.

We acknowledge the limitations of the study and agree with the reviewer's recommendation to consider different pipe types for more comprehensive evaluations. In particular, we have highlighted the need for further experimental studies at the end of the Conclusions section.

Comment 15. The terms rvs and rvd we defined I (8) and (9), but term rs was not defined (Figure 19).

Response 15. Thank you for bringing this to our attention. We apologise for the oversight in not defining the term "rs" before Figure 19. It should have been defined earlier for clarity and consistency. We have now provided a clear definition of "rs" before Figure 19.

Comment 16. Line 512, why the author use the rs (with subscript) and dr (without subscript)?

Response 16. We are grateful for your comments on this point. We agree with you that it is important for clarity and understanding that the notation used is consistent throughout the text. For this reason, the expression 'dr' was replaced by 'dr'."

Comment 17. Too many different conclusions. Some are less important - they can be thrown away.

Response 17. Thank you for your valuable feedback. We carefully reviewed the findings and highlighted the most important ones in the revised manuscript.

Comments on the Quality of English Language

  • English language requires improvement.

Thank you for highlighting this concern. We appreciated your feedback concerning the quality of the English writing. We took your suggestion seriously and took professional assistance to ensure the manuscript met the required standards for publication in a journal.

Reviewer 2 Report

Comments and Suggestions for Authors

General Comment:

Overall, this is a good paper on pipe-soil interactions using experimental and numerical modelling. In my opinion, the paper is interesting and deserves deep attention. I went through it deeply. I can recommend this manuscript be accepted for publication in such journal after doing some required revisions. Some specific comments are given below:

1-       I believe that the presentation of the introduction has some limitations in the current state. Terzaghi's arching theory assumption as well as the Trapdoor test should be introduced with some details so the full idea of Figure 1 can be more clear.

2-       Regarding the experimental setup section, I recommend giving more details about how you calibrated the used earth pressure transducers. Those transducers are so sensitive to temperature at magnitudes beyond the factory specifications. So more details about how you calibrated those sensors to minimize that effect.

3-       For the numerical analysis section, Please add a schematic diagram showing the used Finite element mesh, dimensions, and boundary conditions for the numerical simulation.

4-       How do you calibrate soil parameters mentioned in Table 1? If you used the soil test function in Plaxis, please give more details and show some verification results.

5-       General comments regarding all the results figures from Figure 5 until Figure 23, please replace the word test with the word measured and the word analysis with the word computed. As it is well known any experimental results are measured data and any numerical analysis results are computed results. This will protect the readership of readers and remove any possible confusion. Please modify all figures to be measured and computed.

6-       English writing quality is low and must be improved. Polishing the English with native speaker is recommended.

7- The discussion session is lacking, limitations and recommendations of this research should be highlighted in more details.

8-       Please, try also to improve the conclusion part as it seems to be difficult to understand: Why could it be important this research? What will it be its scientific contribution in future application?

Comments on the Quality of English Language

English writing quality is low and must be improved. Polishing the English with native speaker is recommended.

Author Response

Manuscript ID: applsci-2862955

Murat Gulen1, Havvanur Kilic2

1 Department of Civil Engineering, Yildiz Technical University, [email protected]

2 Department of Civil Engineering, Yildiz Technical University, [email protected]

We sincerely appreciate the time and effort you invested in reviewing our manuscript. It's an honor for us to have the opportunity to submit a revised version. Your feedback and insightful comments have been invaluable to us. We have successfully incorporated changes to address most of the suggestions provided, and these revisions have been clearly highlighted within the manuscript.

Below, we have provided a detailed point-by-point response to your comments and concerns, with corresponding revisions highlighted in red within the resubmitted files. We believe these revisions adequately address your concerns and align with the standards of the journal

Once again, we extend our gratitude for your valuable feedback and constructive criticism.

Murat GULEN (Phd.Candidate)

Yildiz Technical University

Department of Civil Engineering

Reviewer-2

Comment 1. I believe that the presentation of the introduction has some limitations in the current state. Terzaghi's arching theory assumption as well as the Trapdoor test should be introduced with some details so the full idea of Figure 1 can be more clear.

Response 1. Thank you for your valuable feedback on the Introduction section of the manuscript. In response, we have revised the Introduction to provide a more detailed explanation of Terzaghi's arching theory assumption and the trapdoor test, with the aim of improving the reader's understanding of the context. (line 64-71).

Comment 2. Regarding the Experimental Setup part, I recommend giving more details about how you calibrated the used earth pressure transducers. Those transducers are so sensitive to temperature at magnitudes beyond the factory specifications. So more details about how you calibrated those sensors to minimize that effect.

Response 2. The load cells used for stress measurements have a maximum operating temperature of 60°C and a minimum operating temperature of -20°C. The laboratory temperature during the test remained within the operating range of the load cells, minimising the effect of environmental temperature on their accuracy. In addition, as the load cells were placed within and in direct contact with the backfill material, the influence of temperature variations was further reduced, aided by the low thermal conductivity of the backfill material.

In terms of calibration, although the load cells were supplied with a calibration certificate covering specified temperature ranges, we opted for a more accurate calibration method. Each load cell was placed in a steel box and compacted with backfill material in a controlled manner. Once compacted, the steel box was gradually subjected to a uniformly distributed load from above. The millivolt values corresponding to the applied loads were then recorded by the load cells. The calibration steps were completed using the stresses corresponding to the millivolt values and the results were found to be in good agreement with those stated on the calibration certificate. Information on these calibration steps is presented in Section 2: Experimental Setup (Line 119-125).

Comment 3. For the numerical analysis section, Please add a schematic diagram showing the used Finite element mesh, dimensions, and boundary conditions for the numerical simulation.

Response 3. Thank you for your feedback regarding the use of a schematic diagram to enhance the clarity and visual presentation of the numerical analysis section in the text. The manuscript is currently 24 pages and includes 23 figures. We sincerely apologise for not being able to include the requested figures. However, details of these figures for the P test and S test are provided below.

Figure 1. the numerical analysis section a) mesh for P test b) model of finite element for P test c) mesh for S test d) model of finite element for S test

Comment 4. How do you calibrate soil parameters mentioned in Table 1? If you used the soil test function in Plaxis, please give more details and show some verification results.

Response 4. Thank you for your question regarding the calibration of the soil parameters listed in Table 1. One of the main purposes of the S test conducted in this study was to calibrate the parameters of the backfill material, as explained in Section 4.1 and line 267-271.

The Eoed and Eur calculated from the E50 and E50 for the backfill material were obtained from triaxial (CD) tests. The E50 calculated from these tests is approximately 24 MPa. The S test, carried out under localised loading conditions, measured a surface settlement of 8.76 mm at 320 kPa vertical stress. Although the modulus of elasticity derived from the triaxial (CD) tests was used in the analyses performed to validate the S test in Plaxis 2D, repeated analyses showed that the most compatible values of surface settlement from both test and analysis were achieved using the values given in Table 1 (Figure 5).

Comment 5. General comments regarding all the results figures from Figure 5 until Figure 23, please replace the word test with the word measured and the word analysis with the word computed. As it is well known any experimental results are measured data and any numerical analysis results are computed results. This will protect the readership of readers and remove any possible confusion. Please modify all figures to be measured and computed.

Response 5. Thank you for your valuable comments. Throughout the text, we have been careful to use the words "measured" for "data obtained from tests" and "computed" for "data determined from analyses". We have corrected the words that we overlooked and used incorrectly.

Comment 6.  English writing quality is low and must be improved. Polishing the English with native speaker is recommended.

Response 6. We have carefully considered the reviewer's suggestions for improving the quality of the English and the overall readability and clarity of the article. We have therefore made revisions throughout the text with the help of a professional assistant. The reviewer will be able to see these changes in the revised version of the manuscript. These revisions are shown in red throughout the manuscript to indicate the changes made in response to the reviewer's feedback

Comment 7.  The discussion session is lacking, limitations and recommendations of this research should be highlighted in more details.

Response 7. Thank you for your feedback on the Discussion section of the manuscript.  In line with your suggestion, the following explanations have been added to the Discussion part while the recommendations of this study are expressed at the end of the Conclusion part. For example;

However, the preferred backfill and pipe properties for buried pipe systems can vary widely depending on the application. Therefore, the following relationships for loading conditions, backfill and pipe properties are valid for this study (lines 377-380).

In particular, K and σcrown were determined under the assumption that the frictional force occurs in vertical planes (Figure 1). The variation of K and σcrown can be studied under different assumptions regarding the formation, boundaries, and configuration of the arching in the central soil prism (lines 310-313).

Comment 8. Please, try also to improve the conclusion part as it seems to be difficult to understand: Why could it be important this research? What will it be its scientific contribution in future application?

Response 8. Thank you very much for your valuable feedback. We agree with reviewer on the importance of summarising the results and focusing on the key findings of the study. We have carefully reviewed the findings, added to them, and summarised or highlighted the most important ones

Comments on the Quality of English Language

English writing quality is low and must be improved. Polishing the English with native speaker is recommended.

We appreciate the constructive feedback and will ensure the language is refined and free from errors in the revised version of the manuscript.

Reviewer 3 Report

Comments and Suggestions for Authors

The paper conducted full-scale laboratory tests and then used FEM to parametrically studied the effects of main factors on the pipe deflection and soil arching. The research is new and the findings of the research may be useful for geotechnical community. The authors should consider the following aspects:

1.      I don’t think current introduction is written in a good way. The authors used [7-13], [16-30], [31-36], [37-48] in the literature review. It is important to select key publications and stately clearly that what’s the current progress, what’s the main findings and what’s the gap of the research. The introduction should be re-structured.

2.      The novelty of the paper is not well introduced in the introduction.

3.      I suggest the authors to check the language of the paper: typos can be observed throughout the paper, eg re-searcher, moreever, etc.

4.      Figure2: the meaning of “L1, L2, L3…” must be given in the figure.

5.      2.3 loading and 2.4 test procedure, it is not clear the loading details (for instance, loading speed, loading value, etc)

6.      A notation list is useful for the readers to understand the paper.

7.      The authors parametric study based on the experimental results. This is good. However, it is not clear that how these results may be used for the practice, which is important to improve the academic value of the paper. The reviewer suggests the authors to add a Discussion section to illustrate the implication of the results to the practice.

8.      In conclusion, the author listed a lot of “observation” or “existing knowledge”, which is not a good “conclusion” for a top journal paper. The authors should summarize the main findings of the research. Significant improvement on the conclusion is needed. 

Comments on the Quality of English Language

 I suggest the authors to check the language of the paper: typos can be observed throughout the paper, eg re-searcher, moreever, etc. 

Author Response

Manuscript ID: applsci-2862955

Murat Gulen1, Havvanur Kilic2

1 Department of Civil Engineering, Yildiz Technical University, [email protected]

2 Department of Civil Engineering, Yildiz Technical University, [email protected]

We sincerely appreciate the time and effort you invested in reviewing our manuscript. It's an honor for us to have the opportunity to submit a revised version. Your feedback and insightful comments have been invaluable to us. We have successfully incorporated changes to address most of the suggestions provided, and these revisions have been clearly highlighted within the manuscript.

Below, we have provided a detailed point-by-point response to your comments and concerns, with corresponding revisions highlighted in red within the resubmitted files. We believe these revisions adequately address your concerns and align with the standards of the journal

Once again, we extend our gratitude for your valuable feedback and constructive criticism.

Murat GULEN (Phd.Candidate)

Yildiz Technical University

Department of Civil Engineering

Reviewer-3

Comment 1. I don’t think current introduction is written in a good way. The authors used [7-13], [16-30], [31-36], [37-48] in the literature review. It is important to select key publications and stately clearly that what’s the current progress, what’s the main findings and what’s the gap of the research. The introduction should be re-structured.

Response 1. Thank you for your valuable feedback regarding the introduction of the manuscript. We believe that providing explanations for the reference notation [7-13], [16-30], [31-36], [37-48] will help to clarify why we use this notation.

A review of the literature [7-13] shows that each study focuses on a different topic and each topic investigated has a different influence on the results. The suitability of different materials has also been investigated, taking into account the stress-strain curve and geometry of highly compressible materials. For example, references [10, 11] examined the use of EPS geofoam panels, while reference [9] considered the use of a U-shaped geofoam wrap. Reference [10] studied the use of highly compressible materials in deep buried concrete pipe systems, while references [8, 13] focused on their use in culverts. Reference [11] focused on their use in flexible pipe systems and reference [9] on their use in a rigid box. A similar application situation applies to the following sections.

 In   shallow buried flexible pipes, the most critical parameters affecting pipe deflection include the depth and stiffness of pipe burial, the properties of backfill surrounding the pipe, the compaction technique and degree, the soil support supplied to the pipe, the location and frequency of load application, the stiffness of the pavement, distribution of stresses around the pipe, etc. [16-30].

The soil arching exists not only in buried pipes but also in various engineering   constructions such as culverts, tunnels, piled embankments, retaining walls, and    foundations. Researchers have studied the formation, stages, boundaries, and configuration of arching within the soil prism under different assumptions [37-48].

In line with your suggestion, we have included only one example from each study that examines different issues in this field. We have also made additions to the introduction, taking into account the shortcomings and research gaps in the literature (lines 89-93).

Comment 2. The novelty of the paper is not well introduced in the introduction

Response 2. Thank you for your valuable contribution to the Introduction part of the article. We agree with you that the novelty of the research should be made clear in the Introduction. Additions have been made to the Introduction to express the novelty of the study and its contribution to the literature (lines 89-93, 94-105).

Comment 3. I suggest the authors to check the language of the paper: typos can be observed throughout the paper, eg re-searcher, moreever, etc.

Response 3. Thank you for bringing these issues to our attention. We agree that it is necessary to edit incorrect words and sentences in the manuscript. We have carefully addressed these by changing "re-searcher" to "researcher", correcting "moreever" to "moreover", and addressing other inaccuracies throughout the text.

Comment 4. Figure2: the meaning of “L1, L2, L3…” must be given in the figure

Response 4. Thank you for your valuable feedback on Figure 2 and the load cells. We thought it would be more useful to explain the abbreviations "L1, L2, L3..." in the text just above the figure, rather than in the figure (lines 110-114).

Comment 5. 2.3 loading and 2.4 test procedure, it is not clear the loading details (for instance, loading speed, loading value, etc)

Response 5. Thank you for your feedback. In response to your comment, we have revised the manuscript accordingly. The explanations of the loading steps, previously in part of 3.3, have now been moved to part of 2.3 (lines 181-185).

Comment 6. A notation list is useful for the readers to understand the paper.

Response 6.   In consideration of your comments, a list of abbreviations has been added at the end of the article to improve readability and simplify the presentation of information.

Comment 7. The authors parametric study based on the experimental results. This is good. However, it is not clear that how these results may be used for the practice, which is important to improve the academic value of the paper. The reviewer suggests the authors to add a Discussion section to illustrate the implication of the results to the practice.

Response 7. Thank you for your comments. In this study, the Results and Discussion section is combined into a single section rather than being presented separately. This integration is aimed at improving the understanding and evaluation of the results. In particular, the Results and Discussion section comprises about 50% of the article. We believe that adding a new Discussion section and evaluating the results separately would make the manuscript more difficult to follow and read. Therefore, instead of providing a new Discussion part, we have made the additions to the Results and Discussion section, and Conclusions considering the reviewers' suggestions (Such as lines 310-313, 379-380).

Comment 8. In conclusion, the author listed a lot of “observation” or “existing knowledge”, which is not a good “conclusion” for a top journal paper. The authors should summarize the main findings of the research. Significant improvement on the conclusion is needed.

Response 8. Thank you very much for your valuable feedback. We agree with you on the importance of summarising the results and focusing on the key findings of the study. We have carefully reviewed the findings and summarised or highlighted the most important ones.

Comments on the Quality of English Language

  • I suggest the authors to check the language of the paper: typos can be observed throughout the paper, eg re-searcher, moreever, etc.

We have carefully considered the reviewer's suggestions for improving the quality of the English and the overall readability and clarity of the article. We have therefore made revisions throughout the text with the help of a professional assistant. The reviewer will be able to see these changes in the revised version of the manuscript. These revisions are shown in red throughout the manuscript to indicate the changes made in response to the reviewer's feedback

Round 2

Reviewer 1 Report

Comments and Suggestions for Authors

The authors have responded to my comments and improved the text sufficiently.

The article is ready for publication.

Comments on the Quality of English Language

Sufficient for publication

Reviewer 3 Report

Comments and Suggestions for Authors

The authors have addressed all my comments. Therefore, the paper can be published in its present form.